# Synergy and antagonism in the integration of BCR and CD40 signals that control B-cell population expansion

Helen Huang [ID] [1,2,3], Haripriya Vaidehi Narayanan[1,2], Mark Yankai Xiang[1,2,3], Vaibhava Kesarwani[1] & Alexander Hoffmann [ID] [1,2 ✉]

## Abstract

**In response to infection or vaccination, lymph nodes must select antigen-reactive B-cells while eliminating auto-reactive B-cells. B-cells are instructed via B-cell receptor (BCR), which binds antigen, and CD40 receptor by antigen-recognizing T-cells. How BCR and CD40 signaling are integrated quantitatively to jointly determine B-cell fate decisions remains unclear. Here, we developed a differential-equations-based model of BCR and CD40 signaling networks activating NFκB. The model recapitulates NFκB dynamics upon BCR and CD40 stimulation, and when linked to established cell decision models of cell cycle and survival control, the resulting cell population dynamics. However, upon costimulation, NFκB dynamics were correctly predicted but the predicted potentiated population expansion was not observed experimentally. We found that this discrepancy was due to BCR-induced caspase activity that may trigger apoptosis in founder cells, unless timely NFκB-induced survival gene expression protects them. Iterative model predictions and sequential co-stimulation experiments revealed how complex non-monotonic integration of BCR and CD40 signals controls positive and negative selection of B-cells. Our work suggests a temporal proof-reading mechanism for regulating the stringency of B-cell selection during antibody responses.**

**Keywords** B-cell Selection; B-cell Signaling; NFκB; Mathematical Modeling; Systems Immunology
**Subject Categories** Computational Biology; Immunology

## Introduction

A critical component of antibody generation is the T-dependent (TD) activation of naive B-cells in secondary lymphoid organs, which depends on two distinct signals: B-cell receptor (BCR) engagement by antigen and CD40 stimulation from T-cell help

(Bretscher and Cohn, 1970; Akkaya et al, 2018). Based on clonal selection theory, high-affinity antigen-reactive B cells receive a stronger stimulus and hence proliferate to a greater extent, while low-affinity B cells receive a weaker stimulus and do not proliferate as much (Burnet, 1957; Victora and Nussenzweig, 2022). But a successful response must not only enrich high-affinity antigen-reactive B-cells through positive selection, but also eliminate autoreactive B-cells by negative selection. Thus, the stimuli acting on these B-cells determine the outcome between positive and negative selection.

The BCR is a dual-purpose receptor, functioning both to endocytose antigen and present it to T-cells and to initiate signaling. However, how the signaling functions of the BCR are relevant to the selection of B-cells remains unclear. Prior studies demonstrated that BCR signal transduction was short-circuited during T-dependent (TD) responses (Khalil et al, 2012) and that the CD40 signal alone may be sufficient for B-cell clonal expansion (Shulman et al, 2014), while others suggested that BCR signaling is necessary for the survival and priming of B-cells for their positive selection (Chen et al, 2023). Overall, although BCR and CD40 signaling have been profiled experimentally (Damdinsuren et al, 2010; Akkaya et al, 2018), there is a lack of quantitative understanding of how these two signaling events are integrated within the dynamic sequence of those interactions. A mathematical model is thus needed to understand how the two signals interact and jointly determine the appropriate B-cell survival and proliferation to maintain a balance in positive and negative selection.

Both BCR and CD40 signaling pathways converge on the nuclear factor kappa B (NFκB) signaling system. While BCR stimulation activates the canonical NFκB pathway only transiently, CD40 stimulates both the canonical and non-canonical pathways, resulting in prolonged NFκB activity (Sen, 2006). Mathematical models of the NFκB signaling system have been established (Mitchell et al, 2023) and they have been integrated with models of cell fate decision circuits to recapitulate in vitro B-cell population dynamics resulting from the toll-like receptor ligand CpG, a T-independent stimulus (Shokhirev et al, 2015a; Mitchell et al, 2018; Roy et al, 2019). Another set of models explored the feedback

[1]Signaling Systems Laboratory, Department of Microbiology, Immunology and Molecular Genetics (MIMG), University of California Los Angeles, Los Angeles, USA. [2]Institute for Quantitative and Computational Biosciences (QCBio), University of California Los Angeles, Los Angeles, USA. [3]Bioinformatics Interdepartmental Program, University of California Los Angeles, Los Angeles, USA. ✉E-mail: ahoffmann@ucla.edu

mechanisms within the BCR molecular network, which involves protein kinase C β (PKCβ), CARD containing MAGUK protein1 (CARMA1), transforming growth factor β-activated kinase 1 (TAK1) and IκB kinase β (IKKβ) (Shinohara et al, 2014, 2016; Inoue et al, 2016). However, despite many studies of the CD40 signaling pathway (Dadgostar et al, 2002; Elgueta et al, 2009; Akiyama et al, 2012), there is no mathematical model or quantitative understanding of the dynamics of CD40 signaling. Further, no work has been done to combine the BCR and CD40 receptor signaling knowledge and explore how the two signals combine quantitatively to control NFκB signaling and resulting cell fate decisions such that the B-cell population dynamics in response to T-dependent stimulation may be understood or predicted.

Here, we undertook quantitative studies to develop mathematical models for the receptor activation modules of the BCR and CD40. We then tested the reliability of these models by linking them to established cell survival and cell cycle models for quantitative studies of the B-cell population dynamics in response to BCR and CD40 receptor stimulation. The combined model correctly recapitulated the population dynamics data of B-cells stimulated with either stimulus, but simulating the combined stimulus conditions revealed discrepancies with experimental data. Our investigations revealed an unexpected time-dependent functional antagonism that modules the expected synergy between BCR and CD40 signaling. It is exacerbated by BCR-induced caspase activity that can trigger apoptosis in founder cells, unless NFκB-induced survival gene expression protects B-cells in time. Model-guided sequential co-stimulation studies then revealed how temporal signaling dynamics regulate the control of cell fate decisions that underlie negative vs. positive selection of B-cell clonal expansion.

# Results

## A mathematical model of B-cell signaling during T-dependent stimulus responses

To understand how BCR and CD40 signaling are mechanistically integrated during T-dependent immune responses, we developed a mathematical model of the molecular interaction network that downstream of these receptors. We built upon an established mathematical model of the BCR signaling pathway (Inoue et al, 2016) and formulated a new CD40 model to include key mechanistic features of its known signaling pathway (Elgueta et al, 2009; Akiyama et al, 2012). Both BCR and CD40 pathways culminate in canonical and non-canonical IKK activation, defining a T-dependent B-cell signaling network model that includes 37 mass-action equations (Fig. 1A). We parameterized the T-dependent B-cell signaling network model by adopting parameter values from established models, using half-lives and synthesis rates from biochemical experiments in the literature, or manually fitting to published time course data (Table 1).

Stimulation of B cells with TD ligands activates the multidimeric NFκB signaling system to regulate downstream cell fate response. Therefore, once the receptor model outputs appeared to fit the published activation dynamics of IKK induced by TD stimuli (Shinohara et al, 2016), we connected it to the latest mathematical model of the NFκB signaling network that accounts for the time-

dependent activity of multiple NFκB dimers (Mitchell et al, 2023) (Fig. 1B). We then introduced heterogeneity to the signaling dynamics by sampling parameter sets from parameter distributions (see more details in "Computational modeling of the T-dependent receptor signaling pathway" section of the Methods). To test if the BCR/CD40-NFκB model recapitulates the NFκB dynamics induced by TD stimulation, we stimulated naive B cells for 7, 24, and 48 h with different stimulation conditions—low α-CD40 concentration (1 μg/mL), high α-CD40 concentration (10 μg/mL), and costimulation with high concentration of both α-CD40 and α-BCR (10 μg/mL)—to quantify their NFκB signaling activity by immunoblotting the nuclear fraction for RelA (p65) and cRel level (Fig. EV1A), following live cell normalization (Fig. EV1B).

The optimal range of experimental doses of α-CD40 were chosen based on prior literature that carefully examined the B cell proliferation response to defined α-CD40 concentrations (Rush and Hodgkin, 2001; Turner et al, 2008; Hawkins et al, 2013). The chosen experimental doses were determined to be equivalent to model simulations with 6 nM of stimulus for low dose and 30 nM for high dose. We then undertook a systematic dose-response analysis focusing on NFκB, and found that peak nuclear RelA:p50 (Fig. 1C) and nuclear cRel:p50 (Fig. 1D) reach saturation with the high dose, but that cRel:p52 does not. This indicates that the canonical and non-canonical NFκB activities have differential dose responses in response to CD40 stimulation. Thereby the low dose of α-CD40 allows us to examine how BCR and CD40 signals are integrated in unsaturated NFκB conditions; while the high dose of α-CD40 allows us to test their integration under saturated canonical NFκB condition.

The integrated model recapitulated the amplitude, dose-responsiveness, and speed of RelA and cRel dynamics in response to various stimulation schemes (Fig. 1E–H). For example, in both simulated and experimental data, nuclear RelA level in B-cells stimulated with high and low doses of α-CD40 increased 10- and 5-fold, respectively, after 7 h of stimulation (Fig. 1E,F), and nuclear cRel was induced to around 13- and 4-fold relative to its steady-state level (Fig. 1G,H). The model further captured the steeper gradient of downregulation of nuclear cRel than RelA from 7 h to 24 h after stimulation. Although both model simulation and immunoblot results showed a decrease in nuclear NFκB levels at 24 h after stimulation, in vitro data indicated a slight rebound at 48 h (Fig. 1F,H), whereas the in silico levels continued to decrease (Fig. 1E,G). This discrepancy in late-phase NFκB dynamics could be due to in vitro cells undergoing cell death or proliferation by this timepoint (Fig. EV1B), which this signaling-only model does not account for. In sum, the model was able to recapitulate early B-cell NFκB dynamics in response to TD stimulation, but failed to capture late activity in the absence of accounting for cell fate decisions.

## Combining models of signaling and cell fate decisions

Given that NFκB dynamics are critical determinants of B-cell fate decisions (Shokhirev et al, 2015a; Mitchell et al, 2018; Roy et al, 2019), we next asked whether linking the CD40-NFκB signaling model to cell fate decision models would correctly predict the population dynamics in response to TD stimulation. After connecting these modules (Fig. 2A left), we could simulate the time-dependent dynamics of successive generations of B-cells that

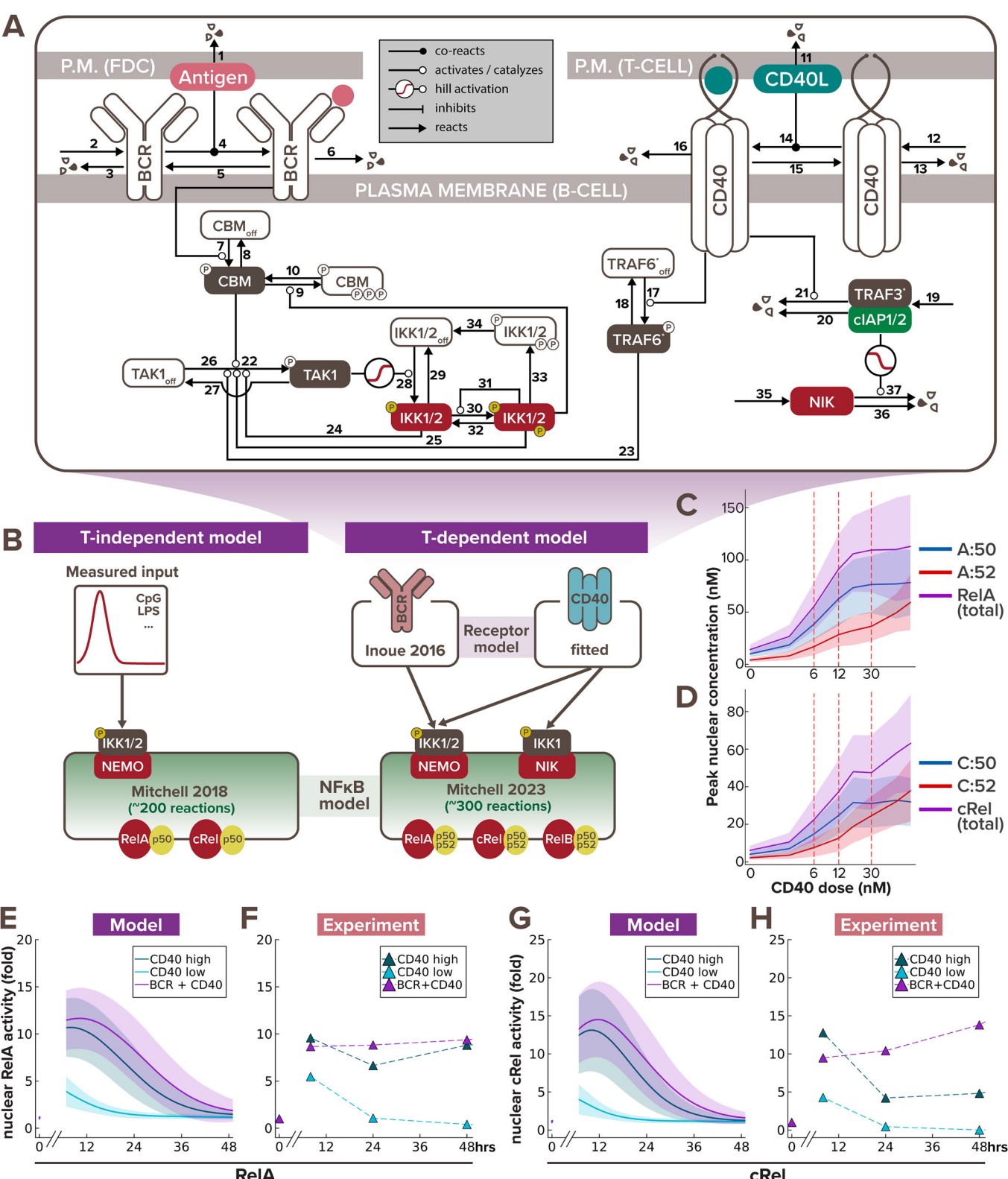

results from division and death decisions. To generate experimental data for comparison, we stained naive B cells with the Cell Trace Far Red (CTFR) dye, cultured them under various CD40 stimulus conditions to observe their proliferation kinetics *via* dye dilution (Fig. EV1C), and quantified the number of cells under each generation using FlowJo's generation deconvolution feature (Roederer, 2011) (Fig. 2A right). Four conditions were used in the experiment and model simulation, respectively: no stimulus, low concentration (1 µg/mL in experiment, which corresponds to 6 nM in model simulation), medium concentration (3 µg/mL/

◀ **Figure 1. B-cell signaling model recapitulates early NFκB dynamics in response to T-dependent stimulation.**

(A) Schematic of BCR-CD40 receptor model to recapitulate T-dependent activation of B-cells. Each of the 37 biochemical reactions capture a process like phosphorylation, dephosphorylation, synthesis, degradation, association, dissociation, or catalysis, and is annotated with a number consistent with the corresponding reaction and parameter value in Table 1. Species with a colored background are in their active form, while those with white background are in their inactive form. Circle with 'P' inside indicates phosphorylation. (B) Schematics of existing T-independent (left) and newly integrated T-dependent (right) NFκB signaling modeling frameworks. T-independent stimulation typically only involves a single ligand (e.g., CpG or LPS), while T-dependent stimulation always involve a more complex receptor signaling system of both BCR and CD40 ligands. Low, medium and high dose of CD40 were set to 6 nM, 12 nM, and 30 nM, respectively, to correspond to the three experimental doses we used (1 μg/mL, 3.3 μg/mL, and 10 μg/mL). (C, D) Line graph from model simulations show (C) peak nuclear RelA (in the form of RelA:p50 and RelA:P52 heterodimers) and (D) peak nuclear cRel (in the form of cRel:p50 and cRel:P52 heterodimers) levels in response to increasing α-CD40 doses, where the shading represents the sample standard deviation of 1000 cells. X-axes are plotted on a log-scale to accommodate a wide range of concentrations. (E) Line graphs from model simulations of 1000 virtual cells and (F) matching experiments with 600 K founder B-cells show temporal trajectories of nuclear RelA level at 0, 7, 24, and 48 h following stimulation with low α-CD40 (1 μg/mL), high α-CD40 (10 μg/mL), or costimulation with high α-CD40 and α-BCR (10 μg/mL each). Darker colored lines represent the average nuclear RelA level from 1000 cells and the lighter shading represents the sample standard deviation of the 1000 cells. (G, H) Line graphs of nuclear cRel level in matching stimulation conditions as (E, F). Source data are available online for this figure.

20 nM), and high concentration (10 μg/mL/60 nM) of α-CD40 stimulus.

Inspecting the data, we found that increasing doses of α-CD40 affect both the time to first division (Tdiv0) and the total number of progeny a naive B-cell can produce, while T-independent (TI) ligands CpG and LPS, which were used in prior studies, show fast Tdiv0 even at low doses; this is consistent with previous observation of a graded response to CD40 stimulation versus a quantal all-or-none response to CpG and LPS (Hawkins et al, 2013; Turner et al, 2008). Our published cell fate module that was tuned to B-cells stimulated with the TI ligand CpG (Shokhirev et al, 2015a; Mitchell et al, 2018; Roy et al, 2019) qualitatively fit the TD ligand CD40 data, but the simulated responses were faster than observed (Fig. EV2A). Meanwhile, the later division times (Tdiv1+) of the CD40 experimental data were shorter than predicted by the model (Fig. EV2A), while the proportion of dividers was lower. To improve the model fit, we identified locally sensitive parameters in the cell cycle module that contribute to Tdiv0 and Tdiv1+ by calculating the standard deviation in division times when scaling each parameter from 0.2 to 5.0 times the original values (Fig. EV2C, see more details in "Local sensitivity analysis to tune CD40-activated cell fates" section of the Methods). After fine-tuning the sensitive parameters, the model appeared to recapitulate key aspects of B-cell population dynamics in response to all tested α-CD40 doses (Fig. 2B,C). For example, the fold change of live cell counts and the proportion of generation 0 cells (non-dividers) relative to generation 1+ cells (dividers) appeared to be consistent between model simulation and experimental results at most timepoints; both features were also captured by the model in a dose-dependent manner.

To quantitatively evaluate the model fit at the population dynamics level, we calculated the root mean square deviation (RMSD) of the population expansion index (Fig. 2D) and generational composition (Fig. 2E) between model and experimental outputs at each experimental timepoint (0, 24, 36, 48, 72, and 96 h). Total RMSDs were evaluated between each model and experiment pair, regardless of matching and mismatching α-CD40 doses. As a permutation null, the dose-mismatched pairs demonstrated high RMSD values of around or above 1.0, while the dose-matched pairs exhibited much lower RMSD values of around or below 0.5 for population expansion index, indicating great fit in all α-CD40 doses (Fig. 2D, right). This marked an improvement from the RMSD values before tuning the cell fate parameters. For example, before tuning, population expansion of B-cells stimulated

with a medium dose in silico had a 1.00 RMSD from its matched in vitro medium dose, higher than its 0.75 RMSD from the mismatched high dose (Fig. EV2B, top). After tuning, medium dose in silico data had a 0.45 RMSD from the matched in vitro medium dose data, much lower than its 1.33 RMSD from the mismatched high dose (Fig. 2D, right). Similar improvement can also be observed by comparing the generational composition RMSD values before (Fig. EV2B, bottom) and after tuning (Fig. 2E, right). In sum, the multi-scale model was able to reliably predict B-cell population dynamics over 96 h in terms of heterogeneous receptor-induced NFκB signaling dynamics and ensuing cell death and division decisions.

## BCR and CD40 costimulation show both synergy and antagonism

As the model captured B-cell population dynamics in response to various doses of CD40 stimulation, we next asked if it can accurately predict the dynamics in response to BCR and CD40 costimulation. We followed the same workflow as in Fig. 2A to generate model simulation and dye dilution data in response to two BCR and CD40 costimulation conditions: first, high α-BCR (10 μg/mL) and low α-CD40 (1 μg/mL) (co-low) and second, high α-BCR (10 μg/mL) and high α-CD40 (10 μg/mL) (co-high). In both cases, the multi-scale model predicted more population expansion in costimulation (Fig. 3A,B) than the corresponding high and low CD40 single-ligand stimulation (Fig. 2B). Because both BCR and CD40 stimuli activate the pro-survival and pro-proliferative NFκB pathway, the model simulation results were consistent with our expectation of synergistic population expansion.

However, experimental results of matching stimulus conditions revealed that the two TD stimuli synergized only in a dose-dependent manner. While the dye dilution data (Fig. 3C) showed a synergistic population expansion in co-low condition as predicted (Fig. 3A), there was an unexpected antagonistic effect of BCR costimulation when combined with high CD40 stimulation (Fig. 3D). Indeed, when we calculated the RMSDs between simulated and experimental data for the 2 costimulation conditions, the co-low condition had a score of 0.93, indicating a good fit that's comparable to the CD40 single-ligand stimulation conditions (that ranged from 0.32 to 0.90), but the co-high condition had a bigger deviation of 1.58, suggesting a poorer fit (Fig. 3E). Notably, the poor RMSD in co-high condition was mainly attributed to the population expansion index, which had an RMSD of 0.89 that's

**Table 1. Receptor model reactions and parameter values.**

| Module | # | Reactions | Rates | Units | Source |
|---|---|---|---|---|---|
| BCR receptor | 1 | ANTIGEN → | 0.05 | $h^{-1}$ | Fitted (antibody) |
| | 2 | → BCR | 4.93 | $nM\,h^{-1}$ | Fitted |
| | 3 | BCR → | 1.43 | $h^{-1}$ | Fitted |
| | 4 | ANTIGEN + BCR → ABCR | 66 | $nM^{-1}\,h^{-1}$ | Fitted |
| | 5 | ABCR → ANTIGEN + BCR | 1.26 | $h^{-1}$ | Fitted |
| | 6 | ABCR → | 0.35 | $h^{-1}$ | Coulter et al, 2018 |
| | 7 | CBM + ABCR → ACBM + ABCR | 6.6 | $nM^{-1}\,h^{-1}$ | Inoue et al, 2016 |
| | 8 | ACBM → CBM | 0.126 | $h^{-1}$ | Inoue et al, 2016 |
| | 9 | ACBM + IKK → ICBM + IKK | 0.181 | $nM^{-1}\,h^{-1}$ | Inoue et al, 2016 |
| | 10 | ICBM → CBM | 0.068 | $h^{-1}$ | Inoue et al, 2016 |
| CD40 receptor | 11 | CD40L → | 0.05 | $h^{-1}$ | Fitted (antibody) |
| | 12 | → CD40R | 7.672 | $nM\,h^{-1}$ | Fitted |
| | 13 | CD40R → | 0.05 | $h^{-1}$ | Tucker and Schwiebert, 2008 |
| | 14 | CD40L + CD40R → CD40LR | 0.04 | $nM^{-1}\,h^{-1}$ | Ceglia et al, 2021 |
| | 15 | CD40LR → CD40L + CD40R | 11.3 | $h^{-1}$ | Ceglia et al, 2021 |
| | 16 | CD40LR → | 0.17 | $h^{-1}$ | Tucker and Schwiebert, 2008 |
| | 17 | CD40LR + TRAF6$_{OFF}$ → CD40LR + TRAF6 | 0.1 | $nM^{-1}\,h^{-1}$ | Cheng et al, 2015 |
| | 18 | TRAF6 → TRAF6$_{OFF}$ | 7.5 | $h^{-1}$ | Cheng et al, 2015 |
| | 19 | → TRAF3 | 10 | $nM\,h^{-1}$ | Fitted |
| | 20 | TRAF3 → | 0.5 | $h^{-1}$ | Zhao et al, 2016 |
| | 21 | CD40LR + TRAF3 → CD40LR | 10 | $nM^{-1}\,h^{-1}$ | Fitted |
| TAK1 dynamics | 22 | ACBM + TAK1 → ACBM + ATAK1 | 1050 | $nM^{-1}\,h^{-1}$ | Shinohara et al, 2014 |
| | 23 | TRAF6 + TAK1 → TRAF6 + ATAK1 | 60 | $nM^{-1}\,h^{-1}$ | Cheng et al, 2015 |
| | 24 | IKK2 + TAK1 → IKK2 + ATAK1 | 401.14 | $nM^{-1}\,h^{-1}$ | Shinohara et al, 2014 |
| | 25 | IKK3 + TAK1 → IKK3 + ATAK1 | 1182.86 | $nM^{-1}\,h^{-1}$ | Shinohara et al, 2014 |
| | 26 | TAK1 → ATAK1 | 249 | $h^{-1}$ | Shinohara et al, 2014 |
| | 27 | ATAK1 → TAK1 | 258600 | $h^{-1}$ | Shinohara et al, 2014 |
| IKK dynamics | 28 | TAK1 + IKK_OFF → TAK1 + IKK2 | 80.72 | $h^{-1}$ | Shinohara et al, 2016 |
| | 29 | IKK2 → IKK_OFF | 2175.6 | $h^{-1}$ | Shinohara et al, 2016 |
| | 30 | IKK2 → IKK3 | 0.009 | $h^{-1}$ | Shinohara et al, 2016 |
| | 31 | IKK3 + IKK2 → IKK3 + IKK3 | 2094 | $h^{-1}$ | Shinohara et al, 2016 |
| | 32 | IKK3 → IKK2 | 53844 | $h^{-1}$ | Shinohara et al, 2016 |
| | 33 | IKK3 → IIKK | 2528.4 | $h^{-1}$ | Shinohara et al, 2016 |
| | 34 | IIKK → IKK_OFF | 957.6 | $h^{-1}$ | Shinohara et al, 2016 |
| NIK dynamics | 35 | → NIK | 12 | $nM\,h^{-1}$ | Mitchell et al, 2023 |
| | 36 | NIK → | 0.231 | $h^{-1}$ | Mitchell et al, 2023 |
| | 37 | TRAF3 + NIK → TRAF3 | 2 | $nM^{-1}\,h^{-1}$ | Qing et al, 2005 |

(Qing et al, 2005; Tucker and Schwiebert, 2008; Shinohara et al, 2014, 2016; Cheng et al, 2015; Inoue et al, 2016; Zhao et al, 2016; Coulter et al, 2018; Ceglia et al, 2021; Mitchell et al, 2023).

much higher than its 0.25-to-0.45 range in CD40 single stimulation conditions (Fig. 2D, right), while the generation composition RMSD was 0.68, only slightly above its 0.06 to 0.53 range in CD40 single stimulation conditions (Fig. 2E, right).

To better understand the source of discrepancy between simulated and experimental population dynamics, we next examined the experimental effects of high α-BCR on the background of CD40 stimulation. High α-BCR costimulation seemed to have a positive effect on B-cells stimulated with a low concentration of α-CD40, increasing both the population expansion index (Fig. 3F) and proliferative capacity (Fig. 3G). The RMSD scores between these two experimental conditions also highlighted that high α-BCR costimulation deviated in both population expansion and generation composition (Fig. 3H). In the context

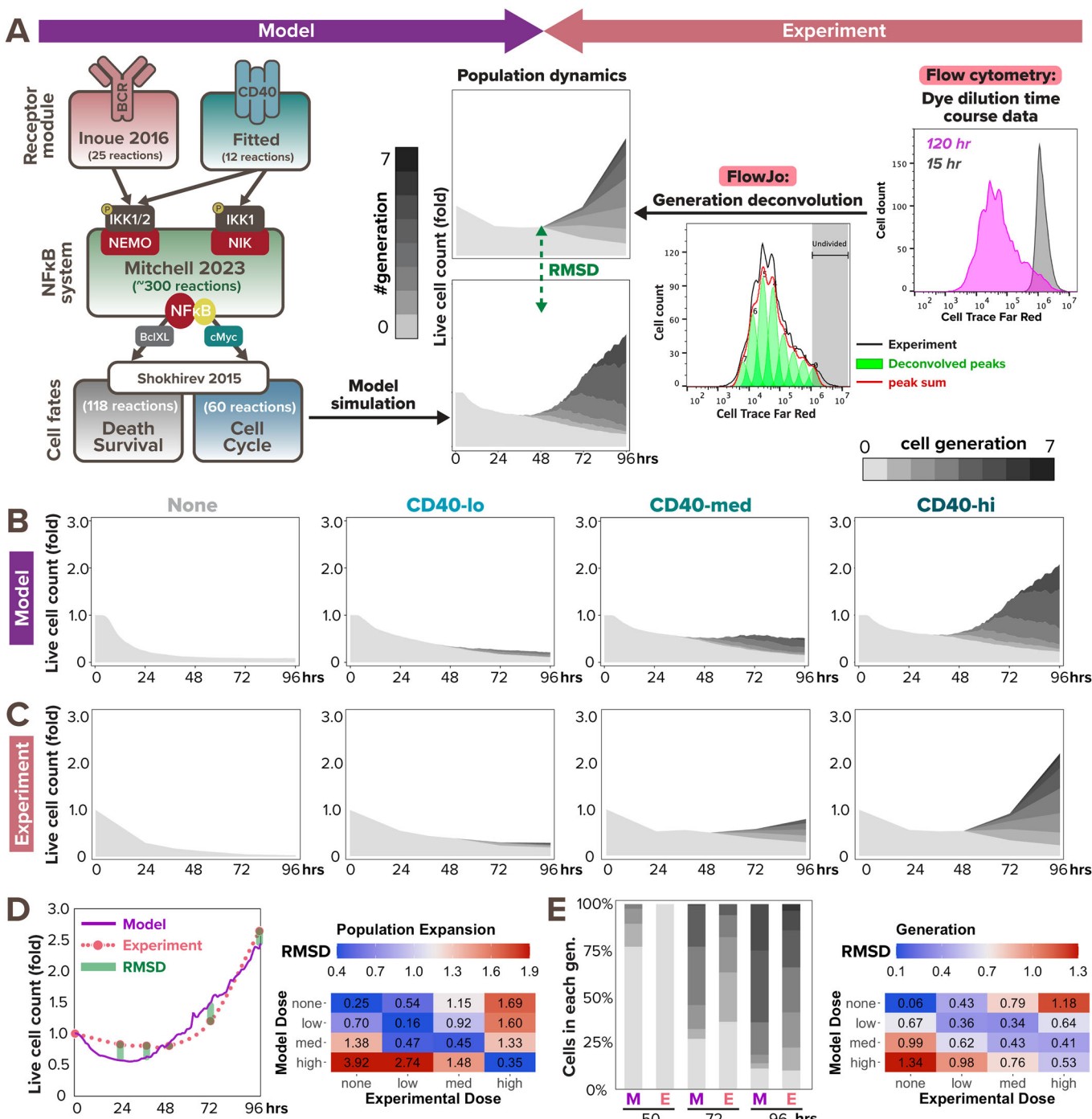

**Figure 2. Multi-scale model recapitulates B-cell population dynamics in response to CD40 stimulation.**

(**A**) Workflow of fitting model simulations to experimental B-cell population dynamics following stimulation. Left: schematic of full T-dependent modeling framework. Right: experimental workflow with Cell Trace Far Red (CTFR) dye dilution. (**B**) Stacked area plots from model simulations of 1000 virtual B-cells show their population dynamics in response to stimulation with (from left to right) no (0 nM), low (6 nM), medium (12 nM), and high (30 nM) dose of α-CD40. Each subsequent generation of proliferating cells is indicated with a darker gray. (**C**) Stacked area plots from matching experiments of 19196 founder B-cells show their population dynamics in response to no (0 μg/mL), low (1 μg/mL), medium (3.3 μg/mL), or high (10 μg/mL) dose of α-CD40. (**D, E**) Root mean square deviation (RMSD) is calculated between simulated and experimental data, and is composed of 2 scores: RMSD of (**D**) relative population size expansion and RMSD of (**E**) generation composition. An example of RMSD between model and experimental data is shown on the left side of (**D**) and (**E**), and a heatmap of the RMSD scores in matching (diagonal) or mismatching (off-diagonal) model-and-experiment pairs is shown on the right side. Lower RMSD scores correspond to better fit. Source data are available online for this figure.

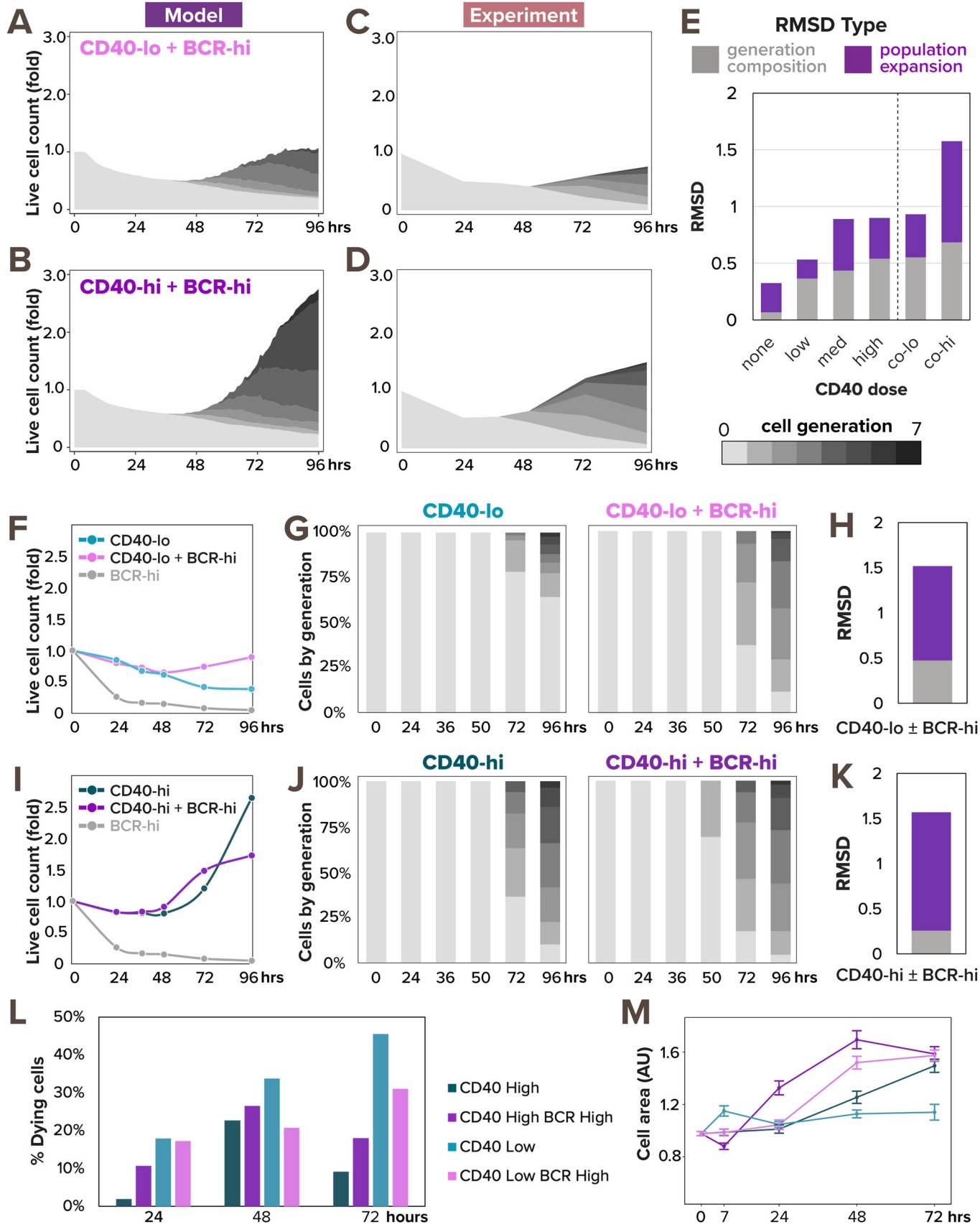

**Figure 3.   Model predicts synergistic population expansion in response to BCR and CD40 costimulation, but experiment reveals dose-dependent interaction between the stimuli.**

(A) Stacked area plot from model simulations of 1000 virtual B-cells show their population dynamics in response to costimulation with high α-BCR (0.25 nM) and low α-CD40 (6 nM). Each subsequent generation of proliferating cells is indicated with a darker gray. (B) Stacked area plot from model simulations of 1000 virtual B-cells show their population dynamics in response to costimulation with high α-BCR (0.25 nM) and high α-CD40 (30 nM). (C) Stacked area plot from matching experiments as (A) of 19196 founder B-cells show their population dynamics in response to high α-BCR (10 μg/mL) and low α-CD40 (1 μg/mL) costimulation. (D) Stacked area plot from matching experiments as (B) of 19196 founder B-cells show their population dynamics in response to high α-BCR (10 μg/mL) and high α-CD40 (10 μg/mL) costimulation. (E) Stacked bar graph shows a breakdown of total RMSD by types in the 2 costimulation conditions compared to the 4 model-and-experiment pairs in Fig. 2B,C which includes no, low, medium, and high dose of α-CD40. (F) Line graph of experimental population expansion index is higher in response to costimulation than without α-BCR. (G) Stacked bar graph of experimental generation composition dynamics in response to low α-CD40 stimulation with or without high α-BCR costimulation. (H) Stacked bar graph of RMSD score between the two experimental conditions in (G) shows the addition of high α-BCR changes both population expansion and generation composition. (I) Line graph of experimental population expansion index is lower in response to costimulation than without α-BCR. (J) Stacked bar graph of experimental generation composition dynamics in response to high α-CD40 stimulation with high α-BCR costimulation. (K) Stacked bar graph of RMSD score between the two experimental conditions in (J) shows the addition of high α-BCR predominantly affects population expansion. (L) Bar graph to compare the percentage of actively dying cells (DRAQ7+ cells, $N = 100–200$ for each bar) in response to 4 stimulation conditions. (M) Line graph of cell area ($N = 100–200$) in the 4 experimental conditions over time, where error bar represents the standard error of the mean (SEM). Source data are available online for this figure.

of high CD40 stimulation, however, the addition of high α-BCR had a less straightforward effect, causing the B-cell population to expand less at 96 h than without α-BCR (Fig. 3I). Conversely, the generation composition chart was similar in the presence or absence of α-BCR (Fig. 3J). The RMSD between these two experimental conditions suggested that the addition of high α-BCR to high α-CD40 altered B-cell population expansion without significantly changing B-cell proliferative capacity (Fig. 3K). High α-BCR thus inflicted opposite effects on the B-cell population dynamics, depending on the context of low or high dose of α-CD40.

To disentangle the dose-dependent synergy and antagonism between BCR and CD40, we did further live cell microscopy studies to observe the effects of high α-BCR stimulation on cell death and cell growth at the single-cell level (Dataset EV1). Using the DRAQ7 viability dye, we quantified the percentage of dead cells over time, and noticed a CD40-dose-dependent BCR effect on cell survival, exacerbating cell death on the background of high α-CD40 but reducing cell death on the background of low CD40 (Fig. 3L). On the other hand, BCR had a CD40-dose-independent effect on cell growth, driving more cell growth compared to stimulation with high or low α-CD40 alone (Fig. 3M).

In sum, the model appeared to accurately predict the NFκB-dependent synergistic signaling interaction between BCR and CD40 at low dose of α-CD40, but failed to reproduce an NFκB-independent antagonistic interaction at co-high dose.

## Considering BCR-induced apoptosis and NFκB saturation

Since the model did not accurately predict the population dynamics in response to BCR-CD40 costimulation, we searched for a mechanistic explanation. Population expansion is a result of both cell proliferation and cell survival, and BCR stimulation (through α-BCR or in vivo antigen) can have pro-proliferative effects on B-cells (Shokhirev and Hoffmann, 2013; Chen et al, 2023). Therefore, the reduced population expansion in the high co-stimulation condition seemed to suggest that α-BCR stimulation had an NFκB-independent anti-survival effect that overrides its pro-survival effect through NFκB signaling. Indeed, it was reported that ligation of the BCR induces cell death in some B cells (Chen et al, 1999; Graves et al, 2004) due to activation of Bcl-2 Interacting Mediator of cell death (Bim) (Gao et al, 2012), caspase-2 or -8 (Chen et al, 1999), mitochondrial dysfunction (Akkaya et al, 2018),

or other pathways. Based on the signaling mechanisms that may mediate activation-induced cell death (AICD) and the available species in the existing cell death module, we revised the T-dependent multi-scale B-cell model to include a simplified pathway from activated BCR to caspase-8 processing (Fig. 4A, see more details in "Computational modeling of BCR-induced cell death" section of the Methods). This processing of pre-caspase-8 into caspase-8 then triggers B-cell death by initiating the cleavage of downstream effector caspases in the cell death module.

To test if the revised model could capture the population dynamics in costimulatory conditions, we re-simulated the virtual B-cell population. With the addition of BCR-induced caspase processing, the simulated cell population in response to co-high stimulation exhibited more cell death and resulted in an overall reduction in population expansion (Fig. 4B, left), which is more consistent with the experimental data (Fig. 4B, right). Meanwhile, Fig. 4C shows slightly less synergy in co-low than previously predicted (Fig. 4A), resulting in more concordance with experimental data as well. A decreased RMSD further confirmed the improvement in model fit (Fig. 4D). Overall, this indicates that the functional antagonism may be mediated by BCR-induced caspase activity triggering apoptosis in founder cells.

To further test the model in which BCR stimulation is pro-proliferative due to NFκB signaling and pro-apoptotic due to AICD, we then asked why the two TD stimuli manifested synergy at co-low stimulation but exhibited antagonism at co-high stimulation (Fig. 4E–H). We simulated nuclear RelA and cRel dynamics, this time with the involvement of cell fate states, and noticed a stronger early NFκB signaling synergy in the co-low condition (Fig. 4E) than the co-high stimulation (Fig. 4F). We next conducted both live cell microscopy and immunoblot experiments to test the model prediction of this discrepancy in early NFκB signaling synergy, using B-cells from a previously developed cRel[mTFP] RelA[mVenus] H2B[mCherry] triple reporter mouse (Fig. EV3, Dataset EV1) (Vaidehi Narayanan et al, 2024). In the context of high CD40 stimulation, the additional high BCR costimulation indeed had minimal effects on nuclear RelA and cRel levels in the first 24 h (Fig. 4H). This lack of early synergy suggested NFκB signaling saturation in high α-CD40 stimulation, such that BCR signal cannot contribute more. On the other hand, in B-cells stimulated with the low, sub-saturating α-CD40 dose, α-BCR amplified nuclear RelA and cRel levels in early times (Fig. 4G). Interestingly, both model simulation

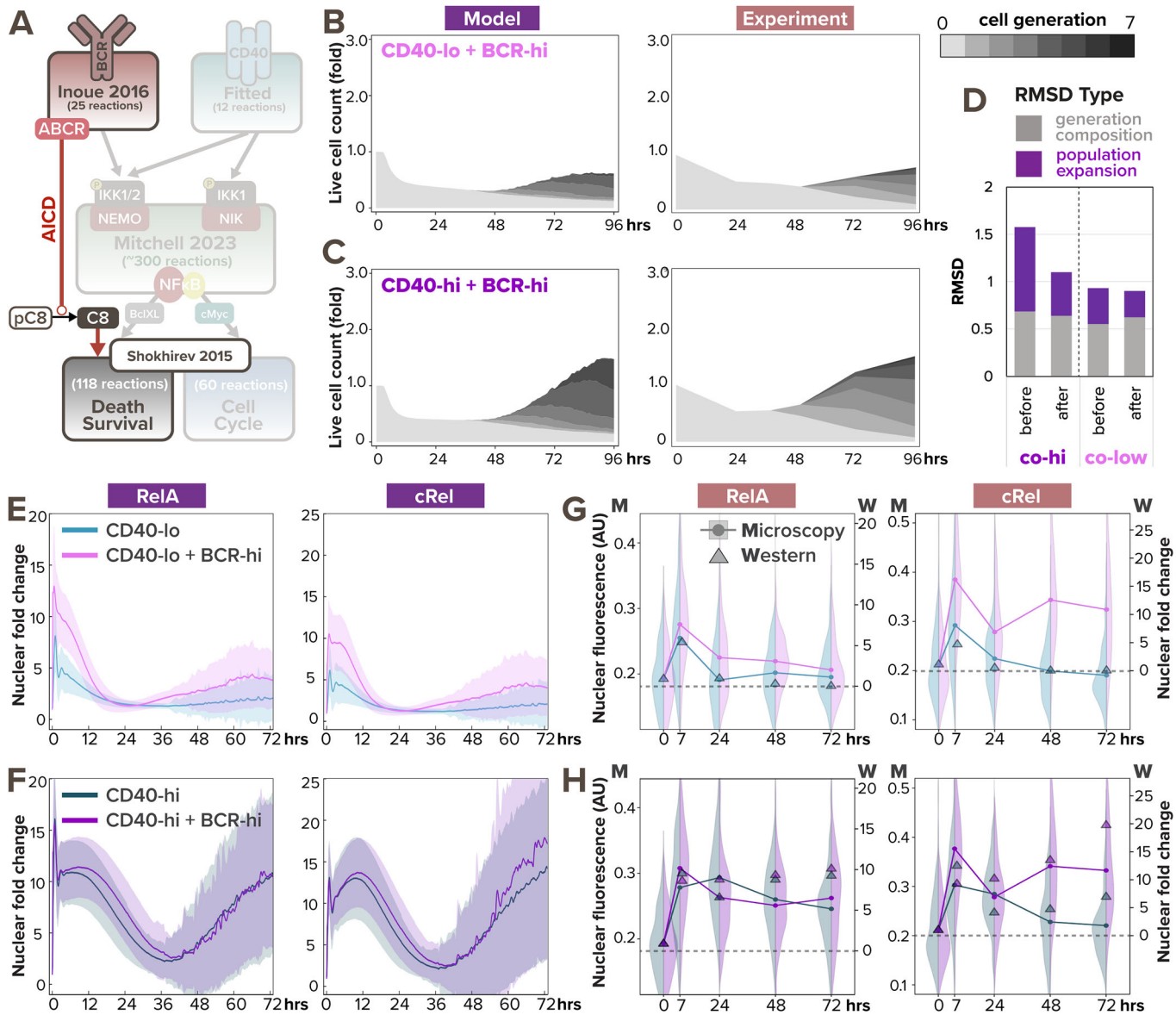

**Figure 4. BCR-induced caspase-dependent apoptosis and NFκB signaling saturation explains the dose-dependent interaction in costimulation.**

(A) Schematic of updated T-dependent multi-scale B-cell model where activated BCR induces caspase-8 processing and subsequently triggers cell death (reaction highlighted in red). (B) Stacked area plots from model simulation (left) and matching experiment (right) show B-cell population dynamics in response to costimulation with high α-BCR (0.25 nM and 10 μg/mL) and high α-CD40 (30 nM and 10 μg/mL) with the addition of BCR-induced caspase processing. Each subsequent generation of proliferating cells is indicated with a darker gray. (C) Stacked area plots from model simulation (left) and matching experiment (right) show B-cell population dynamics in response to costimulation with high α-BCR (0.25 nM and 10 μg/mL) and low α-CD40 (6 nM and 1 μg/mL) with the addition of BCR-induced caspase processing. (D) Bar graph of total RMSDs of the 2 costimulation conditions after the addition of BCR-induced caspase processing compared with before the addition. (E) Line graphs with sample standard deviation (SD) from model simulations show the model-simulated distribution of nuclear RelA (left) and cRel (right) dynamics in response to stimulation with low α-CD40 (light blue) or costimulation with low α-CD40 and high α-BCR (pink). (F) Line graphs with sample standard deviation (SD) from model simulations show the model-simulated distribution of nuclear RelA (left) and cRel (right) dynamics in response to stimulation with high α-CD40 (dark green) or costimulation with low α-CD40 and high α-BCR (purple). (G) Violin plots with corresponding line graphs ($N = 100–200$) connecting the means from multi-channel fluorescence microscopy (left axis) and triangles ($N = 600$ K) from Western blot fold-change quantification (right axis) show the experimental distribution of nuclear RelA and cRel dynamics in response to matching condition as (E). Western blot fold-change values were min-max normalized to be on the same scale as microscopy fluorescence. (H) Violin plots with corresponding line graphs ($N = 100–200$) connecting the means from multi-channel fluorescence microscopy (left axis) and triangles ($N = 600$ K) from Western blot fold-change quantification (right axis) show the experimental distribution of nuclear RelA and cRel dynamics in response to matching condition as (F). Source data are available online for this figure.

and experimental data showed CD40 and BCR signaling in B-cells synergized to potentiate cRel activity (Fig. 4E–H, right), much more than RelA (Fig. 4E–H, left), at later times, regardless of α-CD40 dose. This may be due to positive regulation through autoinduction of cRel (Hannink and Temin, 1990) and the fact that cRel-enriched cells are more likely to proliferate and increase their representation in the population. In sum, the combination of AICD and NFκB signaling saturation explained the dose-dependent interaction in BCR-CD40 costimulation. These results also demonstrated the model's capacity to capture both early and late B-cell NFκB dynamics in response to TD stimuli when simulations account for cell death and proliferation decisions.

## BCR-induced apoptosis can override BCR-induced population growth

As the BCR has the potential to activate both pro-survival signaling via NFκB and anti-survival via caspase-8, we next examined the response relationships of these two pathways and whether the net outcome may be dose-dependent. We first validated that the simulated population dynamics fit the experimental data for costimulation with high dose of α-CD40 combined with three doses of α-BCR (Fig. 5A). We found consistent population dynamics in experimental and simulation studies, in which the dose of BCR stimulus had only subtle effects. We observed that in both model-simulated and experimental populations, the number of non-proliferating cells (lightest gray) is the lowest at 96 h when costimulated with high α-BCR, while the proliferating cell populations (darker grays) remain comparable across conditions (Fig. 5B). This indicates that when costimulated with a high α-CD40 dose, cell survival and overall population expansion monotonically decrease with increasing α-BCR dose from zero, low, to high.

We next focused on how these stimuli potentially affect cell survival which could censor the proliferation module in the multi-scale model. Simulating B-cells stimulated with various doses of either α-BCR or α-CD40, we observed distinct dose-response patterns. For CD40, the higher the dose, the shallower the Kaplan–Meier survival curve is, indicating a monotonic pro-survival effect of CD40 stimulation (Fig. 5C). On the other hand, the survival dose response to BCR stimulation appeared non-monotonic (Fig. 5D), with low-BCR-stimulation increasing the probability of survival over unstimulated cells, but high-BCR-stimulation actually reducing the probability of survival. When we quantified the number of surviving cells in the first 24 h (Fig. 5E), we clearly observed the difference between the two distinct dose-response patterns in terms of monotonicity for CD40 vs BCR agonists.

To gain a systematic understanding of the effects of BCR-induced apoptosis on B-cell response in all combinations of α-BCR and α-CD40 doses, we simulated 25 single or costimulation scenarios, each with 1000 founder B-cells. We then used locally estimated scatterplot smoothing (LOESS) to fit a smooth curve through this scatterplot of 25 data points to generate heatmaps of cell survival rate, proliferation capacity, and population fold-change (Fig. 5F–N). Without AICD, we observed monotonic increase in all metrics with respect to both α-BCR and α-CD40 doses (Fig. 5F–H). When we incorporated AICD in the model, all the metrics still monotonically increased with respect to increasing α-CD40 doses

(Fig. 5I–K), and the divided cells percentages remained unchanged (Fig. 5G,J) with little difference in the percentage dividers (Fig. 5M), as expected. However, with increasing α-BCR doses at a low (or zero) α-CD40 dose, the cell survival rate first increases then decreases (Fig. 5I,L, left 2 columns). When the α-CD40 dose is medium or high, increasing doses of α-BCR monotonically decreased the cell survival rate (Fig. 5I, right 3 columns), due to AICD reducing 24 h cell survival rate by 30% in high dose of α-BCR (Fig. 5L). Examining the resulting population fold-change (Fig. 5H,K), we observed that BCR-induced apoptosis prevented BCR stimulation from promoting population growth, and rendered the B-cell response independent of BCR signaling.

## A temporal window of opportunity to acquire CD40 signals

In TD activation, a B-cell first experiences a BCR signal when binding the antigen, and then a CD40 signal when it has found a T-cell that also recognizes the antigen (Bretscher and Cohn, 1970; Parker, 1993). The time delay between the two signals is determined by the B-cell searching for T-cell help (Okada et al, 2005). Given that our multi-scale model captured B-cell NFκB and population dynamics in response to all tested doses of CD40 stimulation as well as BCR-CD40 costimulation doses, we asked if it could provide some insights on how the two signals combine in the more physiological TD stimulation scenario of sequential BCR and CD40 stimulation (Fig. 6A–C). To simulate this scenario, a one-hour BCR signal was initiated at 0 h, and the CD40 signal was initiated at 1, 3, 5, or 8 h. In this stimulation scenario using high α-BCR + low α-CD40, the multi-scale model predicted that the B-cell population decreases over time, with a steeper decrease when the gap is longer (Fig. 6B, left). In contrast, in the low α-BCR + high α-CD40 sequential stimulation condition, the B-cell population increased drastically after 48 h despite an initial decrease in the first 24 h. That initial decrease was faster when the time gap was 8 h than 1 h gap but the resulting population size at 96 h was similar (Fig. 6B, right). However, when we simulated the high α-BCR + high α-CD40 stimulation scenario, the model simulations exhibited a larger variation in population size at 96 h, where an 8 h gap resulted in less than half the number of live cells than a 1 h or 3 h gap (Fig. 6B, middle). This indicated that there is a limited temporal window of opportunity for B-cells to acquire CD40 signal that rescues a crashing cell population following BCR stimulation.

To test these model predictions, we undertook experiments with these stimulation scenarios. We stained naive B cells with the Cell Trace Far Red (CTFR) dye, stimulated them with low (1 μg/mL) or high (10 μg/mL) dose of α-BCR for 1 h, washed, and cultured them under low (1 μg/mL) or high (10 μg/mL) α-CD40 at 1, 3, 5, or 8 h after α-BCR pre-activation for 4 days (Fig. 6A). We then observed and analyzed their proliferation kinetics via dye dilution using the same workflow as in Fig. 2A. Experimental results demonstrated distinct effects of time-gaps on B-cell population dynamics across sequential stimulation conditions (Fig. 6C). Specifically, increasing the time gaps between BCR and CD40 stimulation had relatively small effects on B-cell population fold-change in both high α-BCR+ low α-CD40 (Fig. 6C, left) and low α-BCR + high α-CD40 sequential stimulation (Fig. 6C, right), as the colored lines representing different time-gaps followed each

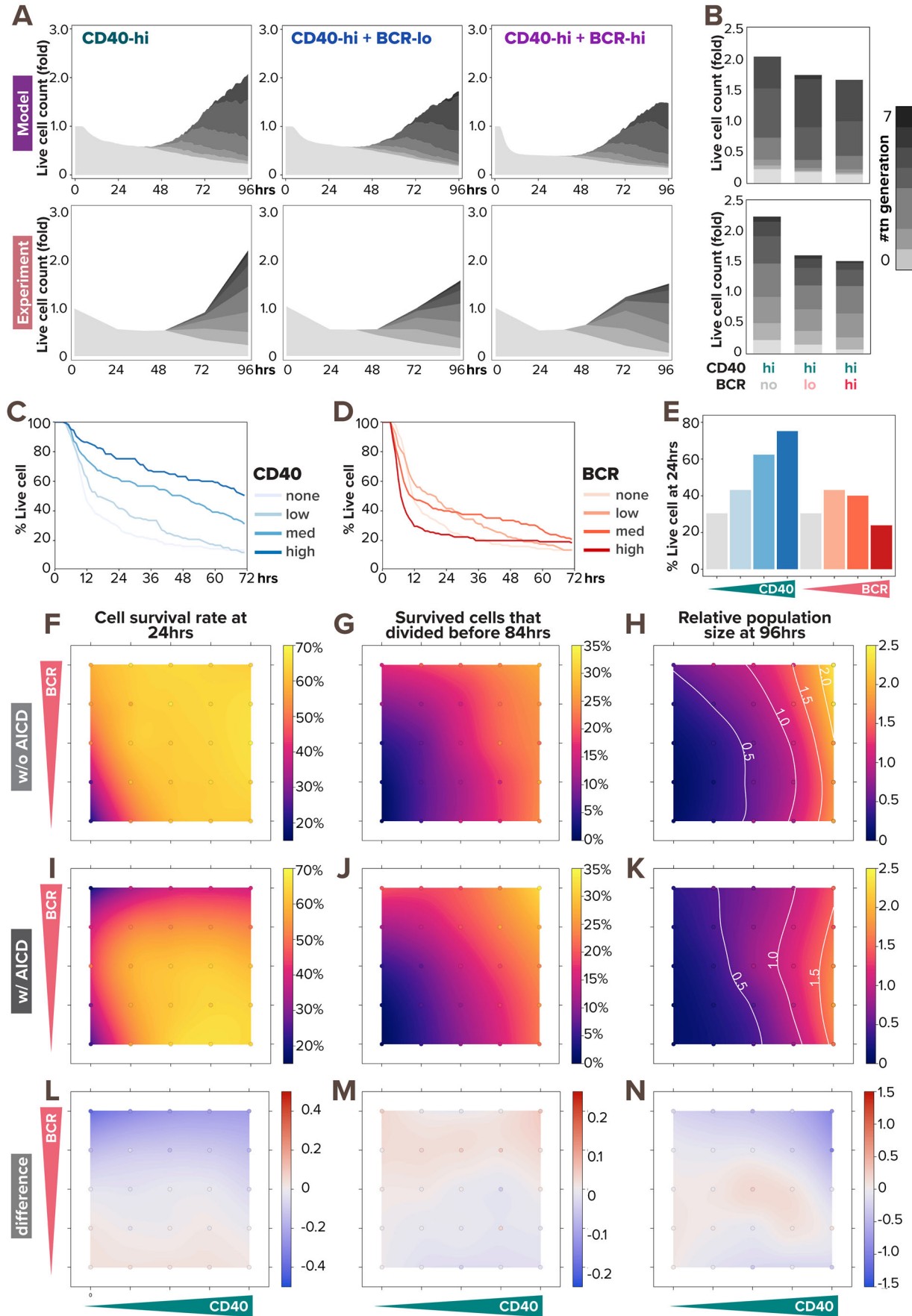

Figure 5. BCR-induced apoptosis prevents BCR stimulation from promoting population growth.

(A) Stacked area plots from model simulations of 1000 virtual B-cells (top) and matching experiments of 19196 founder B-cells (bottom) show their population dynamics in response to costimulation with high (30 nM and 10 μg/mL) α-CD40 and no (0 nM and 0 μg/mL), low (0.005 nM and 1 μg/mL), or high (0.25 nM and 10 μg/mL) dose of α-BCR under the impact of AICD. Each subsequent generation of proliferating cells is indicated with a darker gray. (B) Stacked bar graph from model simulations (top) and experiments (bottom) show a breakdown of live B-cells by generation numbers at 96 h post-stimulation-onset. (C, D) Model-simulated Kaplan–Meier survival curve in response to (C) α-CD40 dose and (D) α-BCR dose shows distinct pattern regarding monotonicity. (E) Bar graph from model simulations show percentage live B-cells at 24 h in response to increasing α-CD40 and α-BCR doses. (F–K) Heatmaps from model simulations of 1000 virtual B-cells in response to 25 single- or co-stimulation scenarios (with 5 doses of α-CD40: 0, 6, 12, 18, and 30 nM, and 5 doses of α-BCR: 0, 0.0005, 0.005, 0.05, and 0.25 nM, combinatorially) show the percentage of survived B-cells at 24 h under (F) no AICD and (I) with AICD, the percentage of proliferative B-cells by 84 h out of those that survived (G) without AICD and (J) with AICD, and the relative population size at 96 h (normalized to founder cell population size) (H) without AICD and (K) with AICD, where white contour lines represent 0.5-, 1.0-, 1.5-, and 2.0-fold changes. (L) Heatmap shows the differences between (F) and (I). (M) Heatmap shows the differences between (G) and (J). (N) Heatmap shows the differences between (H) and (K). In (F–N), 25 simulated doses are plotted as colored circles in a scatterplot, whereas the space in between doses is interpolated with a locally estimated scatterplot smoothing (LOESS) curve. Source data are available online for this figure.

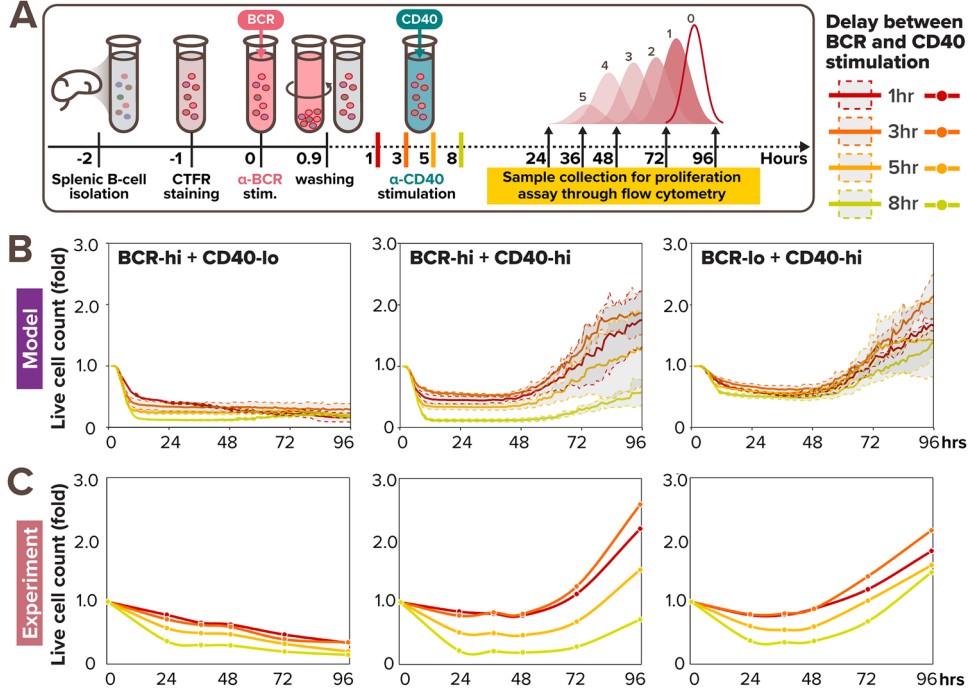

Figure 6. Sequential BCR-CD40 simulation reveals a limited window of opportunity to acquire CD40 signal.

(A) In vitro experimental workflow where primary B cells are sequentially stimulated with pulsing α-BCR, followed by α-CD40 stimulation 1, 3, 5, or 8 h later. (B) Line graph from model simulations of 1000 virtual B-cells show their population expansion in response to sequential costimulation with high α-BCR (0.25 nM) and low CD40 (6 nM) (left), high α-BCR (0.25 nM) and high α-CD40 (30 nM) (middle), and low α-BCR (0.005 nM) and high α-CD40 (30 nM) (right), colored by the gap between α-BCR and α-CD40 stimulation. Each thick colored line represents the average population expansion from 1000 cells, and the shading represents the population standard deviation from the 8 simulations, each with 125 founder cells. (C) Line graph from matching experiments of 19196 founder B-cells show their population expansion at 24, 36, 48, 72, and 96 h in response to sequential costimulation with high α-BCR (10 μg/mL) and low α-CD40 (1 μg/mL) (left), high α-BCR (10 μg/mL) and high α-CD40 (10 μg/mL) (middle), and low α-BCR (1 μg/mL) and high α-CD40 (10 μg/mL) (right), colored by the gap between BCR and CD40 stimulation. The lines are smoothed using Excel's "smoothed line" function which used a Catmull-Rom spline. Source data are available online for this figure.

other closely. On the other hand, larger time-gaps (5–8 h) significantly diminished B-cell population compared to smaller time-gaps (1–3 h) in high α-BCR + high α-CD40 sequential stimulation (Fig. 6C, middle). Surprisingly, the time-gap also appeared to have a non-monotonic effect on B-cell population expansion, where cells stimulated 3 h apart (orange) resulted in higher fold-change than cells stimulated 1 h-apart (red), both in silico (Fig. 6B) and in vitro (Fig. 6C, middle & right). This may be due to reduced NFκB signaling saturation when the two stimuli were further apart, while being still close enough to allow for rescue from AICD. In sum, the experimental results were consistent with

the model simulation, confirming the existence of a limited window of opportunity at the high α-BCR + high α-CD40 sequential stimulation regime (Fig. 6C, middle).

Overall, our in silico and experimental investigations of the temporal relationship between these antagonistic signals revealed a limited time window within which CD40 signaling may effectively rescue cell death triggered by BCR signaling. The size of the temporal window depends on the strength of the BCR and CD40 signals, but when the time gap exceeds a threshold of about 5 h, the opportunity to trigger B-cell population expansion is severely diminished.

## Noisy BCR-induced Bcl-xL expression determines the window of opportunity

We next asked what may determine the window of opportunity and the heterogeneous survival outcomes in single B-cells that are a prerequisite for subsequent population expansion. As BCR stimulation was shown to be both pro-survival due to NFκB-induced Bcl-xL activity and anti-survival due to AICD (Fig. 7A) resulting in a non-linear dose response (Fig. 5), we examined how this paradoxical signaling affects single B-cell responses in silico model simulations. In the apoptosis pathway, activation of caspase-8 leads to downstream activation and oligomerization of Bax to the mitochondrial outer membrane, forming Bax pores that trigger mitochondrial outer membrane permeabilization (MOMP). The prominent anti-apoptotic Bcl-xL protects cells from MOMP by sequestrating Bax from oligomerization or by retrotranslocating Bax to the cytosol (Dou et al, 2021).

We first examined Bcl-xL and caspase-8 trajectories in 3 stimulation conditions (Fig. 7B).

Noticeably, a decline in free Bcl-xL level (thin lines) correlated with a substantial increase in caspase-8 activity (denoted by a quick color transition from deep blue to pink), the timing of both corresponds to a decline in cell survival (thick line). While CD40 stimulation induces Bcl-xL in a homogeneous manner (Fig. 7B, left), BCR stimulation introduces more heterogeneity in Bcl-xL level among different cells (Fig. 7B, middle and right). Because Bcl-xL is known to be induced by NFκB transcription factors (Chen et al, 2000; Shokhirev et al, 2015b), we also reported RelA and cRel nuclear activity and found that B-cells with higher NFκB activity induced their Bcl-xL levels faster and to a higher extent (Fig. 7C), protecting the cells until the onset of CD40 stimulation. On the other hand, cells with lower NFκB activity could not generate enough anti-apoptotic Bcl-xL and ceased to live (indicated by discontinued lines). Cells that survived the first 12 h had significantly higher peak RelA, cRel, and Bcl-xL activity than cells that died (Fig. 7D). The variability in BCR-induced nuclear NFκB level was consistent with a previous report (Shinohara et al, 2014), where the TAK1-IKK2 positive feedback resulted in a switch-like behavior in BCR activation. Furthermore, BCR-induced Bcl-xL expression was also found to have a much wider distribution than basal, with only part of the population being upregulated, and a moderately positive correlation with cRel expression (Berry et al, 2020).

To gain a systematic understanding of the effects of the time gap between CD40 and BCR stimulation on B-cell response in all combinations of α-BCR and α-CD40 doses, we again simulated 25 single or sequential scenarios, each with 1000 founder B-cells, to generate maps of cell survival rate, proliferation capacity, and population fold-change (Fig. 7E–J). With a 1 h pulse in α-BCR stimulation, we observed a cell survival trend (Fig. 7E) similar to that with coincident costimulation than the scenarios in Fig. 5J. With an 8 h staggered stimulation (Fig. 7F), the effect of AICD on cell survival was much stronger than in coincident costimulation (Fig. 5J), showing heightened cell death within the first 24 h at medium and high α-BCR doses. The two population size maps in Fig. 7H,I showed very little difference at low α-CD40 doses but demonstrated the biggest differences in the upper and lower right corners, where virtual B-cells were stimulated with high α-CD40 doses and either high or no α-BCR doses (Fig. 7J). Overall,

these results clearly illustrated the importance of not only α-BCR and α-CD40 stimulation doses but their temporal relationships in determining cell fates and ultimately population expansion.

## Discussion

In this work, we investigated how the BCR-mediated signal I and the CD40-mediated signal II are integrated in the B-cell fate decision process to clarify their roles in T-dependent B-cell selection. Prior work demonstrated that BCR and CD40 signaling synergize at the level of NFκB activation (Damdinsuren et al, 2010), but did not determine how these signals combine to determine the subsequent B-cell fate decisions and thus the emergent population dynamics. Here, we presented a mechanistic mathematical model of B-cell signaling and fate decision in response to T-dependent stimulation scenarios that recapitulates experimental observations (Figs. 1–4) and could be used to explore the biological consequences of the dose and temporal relationship between type I and II signals (Figs. 5–7). We showed that while BCR signaling has the potential to prime B-cells for positive selection by synergizing with CD40 on NFκB signaling (Fig. 3A,B,F–H), it could also initiate negative selection by functionally antagonizing CD40 signaling through AICD (Figs. 3 and 4). Our work suggests that BCR signaling is the key to tuning the balance between positive and negative selection in mature B-cells.

To construct a tractable mathematical model, we took a parsimonious approach to abstract the signaling pathway initiated by the T-dependent stimuli leading to NFκB. For example, CD40 ligand engagement recruits adapter proteins, which include several tumor necrosis factor receptor-associated factor (TRAF)s, such as TRAF1, TRAF2, TRAF3, TRAF5, TRAF6, and a combination of their complexes (Elgueta et al, 2009). To avoid the complexity of combinatorial biochemical reactions among the TRAF complexes, we used TRAF3 to represent the TRAF2-TRAF3 complex that constitutively inhibits the non-canonical NFκB pathway, and TRAF6 to represent the TRAF1-TRAF2, TRAF3-TRAF5, and TRAF6-TRAF2 complexes that all activate the canonical NFκB pathway. The construction of the CD40 signaling model further included parameters extracted from a substantial literature of experimental studies. For example, the degradation rate of NIK was calculated from its half-life (3 h) estimated in a pulse-chase assay for B-cells stimulated with BAFF and α-CD40 (Qing et al, 2005). The differential degradation rates of CD40 receptor (CD40R) and ligated CD40R (CD40LR) were obtained from cell surface biotinylation assay at the surface of 9HTEo- epithelial cells (Tucker and Schwiebert, 2008); similar parametrization applied to the internalization rates of BCR and antigen-ligated BCR (ABCR) (Coulter et al, 2018). Furthermore, the rates of association and dissociation between CD40 and α-CD40 were derived from the Ka and Kd values determined by surface plasmon resonance (SPR) binding analysis (Ceglia et al, 2021). Having incorporated substantial molecular details and biochemical data, the model serves as a framework for an in silico laboratory that could be expanded and revised iteratively with wet-lab experiments for mechanistically investigating the effects of various genetic and pharmacological perturbations on T-dependent-activated B-cell population dynamics. For example, this detailed model could allow personalizing simulations by adapting the gene-regulatory network

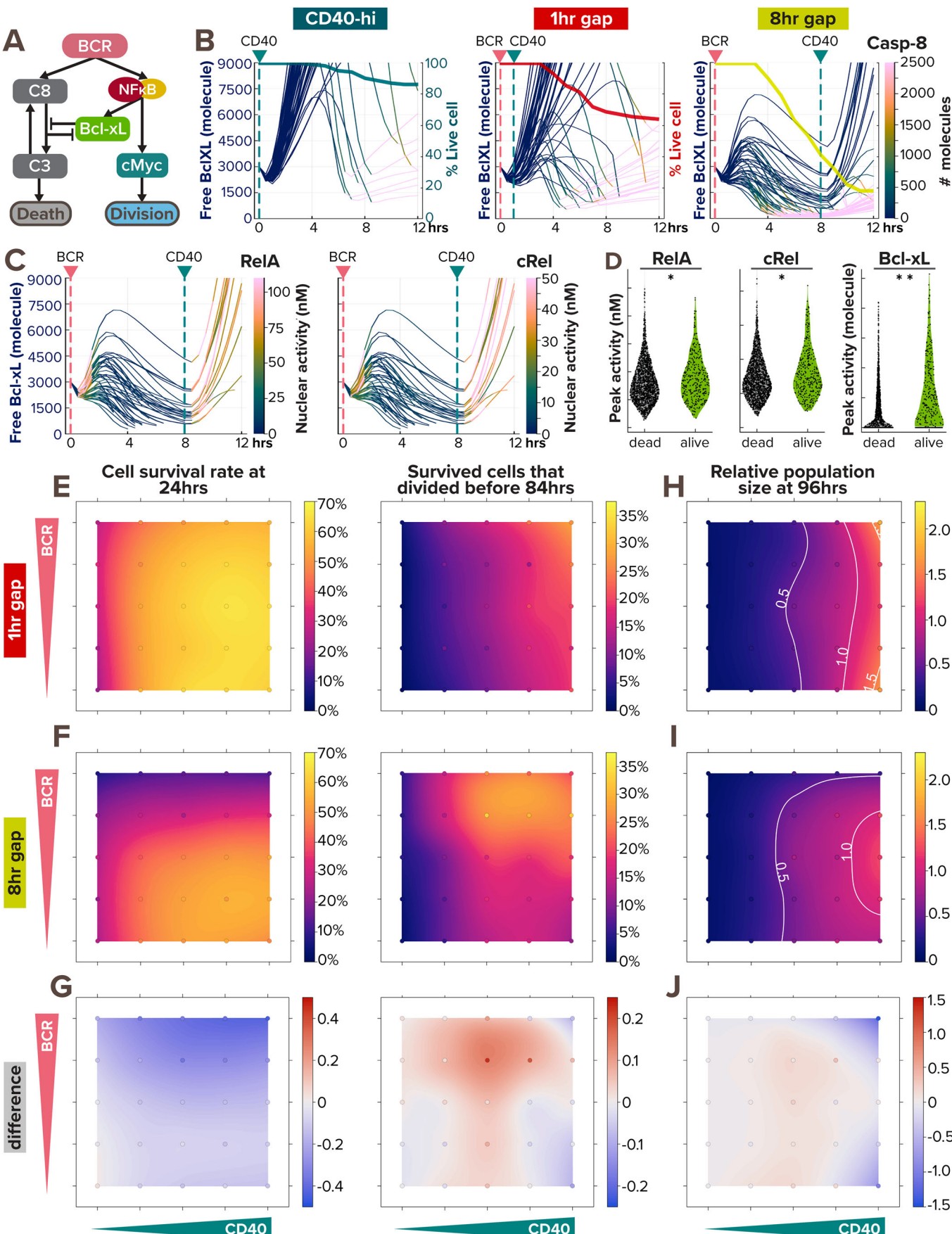

**Figure 7.  BCR-induced anti-apoptotic BclXL protects cells from dying from AICD, as a form of paradoxical signaling.**

(A) Schematic of paradoxical BCR signaling that promotes proliferation, survival, and death of B-cells through cMyc, Bcl-xL, and caspase-8, respectively. (B) Line plots of Bcl-xL activity (left axis) colored by caspase-8 level (color bar) in 50 model-simulated single-cells show the correlation between Bcl-xL consumption and caspase-8 activity. The thick line overlaid on top is a Kaplan–Meier survival curve (right axis). The pink and green vertical dashed lines represent the timing of BCR and CD40 stimulation, respectively. From left to right, the 3 conditions are: high α-CD40 costimulation, and sequential α-BCR and α-CD40 stimulation with a 1 h and 8 h gap. When a cell dies, the line continues (and becomes pink due to high caspase-8 level). (C) Line plots of Bcl-xL activity colored by RelA (left) and cRel (right) activity in 50 model-simulated single-cells demonstrate the correlation between NFκB activation and BclXL upregulation in B-cells costimulated sequentially with an 8 h gap. When a cell dies, the line discontinues. (D) Violin plot of peak RelA, cRel, and Bcl-xL activity in 2000 model-simulated B-cells in response to sequential costimulated with an 8 h gap show the differences between cells that died within the first 12 h and those that survived. Statistical significance is evaluated using Mann–Whitney U tests, with p-values of 0.0019, 0.0077, and <1e−18, correspondingly. Mann–Whitney U test is appropriate because the groups in comparison are not normally distributed but are independent and similar in shapes. (E–J) Heatmaps from model simulations of 1000 virtual B-cells in response to 25 single or costimulation scenarios (with 5 doses of α-CD40: 0, 6, 12, 18, and 30 nM, and 5 doses of α-BCR: 0, 0.0005, 0.005, 0.05, and 0.25 nM, combinatorially) show the percentage of survived cell at 24 h after stimulation onset (left) and the percentage of survived cells that proliferated by 84 h (right) in (E) 1 h sequential costimulation or (F) 8 h sequential costimulation. (G) Heatmap highlights the difference between (E) and (F). (H, I) Heatmap of relative population size at 96 h between (H) 1 h sequential stimulation and (I) 8 h sequential stimulation shows the biggest difference (J) in the upper right and lower right corners, where white contour lines represent 0.5-, 1.0-, and 1.5-fold changes. In (E–J), 25 simulated doses are plotted as colored circles in a scatterplot, whereas the space in between doses is interpolated with a locally estimated scatterplot smoothing (LOESS) curve.

based on genetic variants present in cancer patients for personalizing prognosis (Norris et al, 2024) as well as their individual responses to various pharmaceuticals (Sevrin et al, 2024).

Our previous work identified NFκB as a key determinant of B-cell population dynamics, and quantified the relative contributions of cRel- and RelA-containing NFκB dimers to downstream cell fate effector functions (Shokhirev et al, 2015a; Roy et al, 2019). Here, we present an extended mathematical model, which demonstrated that NFκB-induced survival and proliferation as well as BCR-activation-induced apoptosis are sufficient to explain the survival and proliferation kinetics of B-cells in the explored conditions. Although other signaling pathways that are induced by BCR and CD40, such as PI3K and MAPK, could also play a role, without perturbation studies of these pathways their roles are not quantifiable and are only implicit in the current mathematical model. In response to TD stimuli, the roles of the MAPK p38 and ERK pathways are thought to be minor and generally cooperative with NFκB (Dadgostar et al, 2002).

To fit the temporal dynamics of proliferative responses to TD stimulation, we adjusted a few parameters in the cell cycle model that was adopted from our previous work on TI ligand CpG (Fig. EV2). CD40 generally stimulates a stronger NFκB activity than CpG at saturating doses as CD40 activates also the non-canonical pathway, thereby relieving the IκBδ brake on canonical signaling (Zarnegar et al, 2004; Rodriguez et al, 2024). However, the proliferative response to CpG is faster and stronger, suggesting that another pathway, such as MAPK which is strongly induced by the MyD88 adaptor (Caldwell et al, 2014; Cheng et al, 2017), may be responsible for boosting cell cycle entry in response to some TI stimuli. Our revised model based on the CD40 stimulus that does not activate much MAP p38 pathway is thus more accurate in recapitulating the control of cell growth and cell cycle in response to NFκB.

To test the model predictions, we isolated naive B-cells from mouse spleen for in vitro experimental studies; this allowed us to obtain granular datasets on signaling dynamics and cell fate decisions with dynamic population response. The possibility that B-cells may behave differently in the in vivo lymph node microenvironment (Young and Brink, 2021) could affect the reliability of extrapolating our conclusions from the present study to the in vivo phenomena of positive and negative selection in the germinal center (GC). For example, while naive B-cells are

recruited to undergo affinity maturation (Nowosad et al, 2016), subsequent rounds of selection and proliferation involve GC B-cells (or centrocytes and centroblasts). These have distinct characteristics in signaling: while naive B-cells pick up soluble antigen, GC B-cells uptake antigens that are presented in immune complexes on the surface of follicular dendritic cells (FDCs). GC B-cells also have lower BCR expression and show lower BCR-induced NFκB activation (Young and Brink, 2021), though this may be mitigated by BAFF costimulation from the FDC (Aguzzi et al, 2014). As in naive B-cells, GC B-cell response to CD40 is dominantly regulated by the NFκB signaling pathway (Luo et al, 2018). At the level of proliferative response, GC B-cells undergo cyclic re-entry, hence they may not reach the same division number in a single round as the in vitro assay suggests. For example, a previous study has shown that pre-GC B-cells undergo just 4–6 divisions after the first round of stimulation before becoming GC B-cells (Zhang et al, 2017). Overall, stimulation from the two TD processes (antigen uptake and Tfh help) are expected to synergize at the level of NFκB signaling and proliferative response in both naive and GC B-cells. In addition, administering soluble antigen during an active GC response is highly effective at inducing antigen-specific GC B cell death (Pulendran et al, 1995; Silva et al, 2017), confirming that the BCR-activation-induced death pathway is intact in GC B-cells. Therefore, while the parameters that determine the quantitative aspects of BCR and CD40 synergy and antagonism may differ in naive and GC B-cells, the concept of temporal proofreading mediated by the BCR-CD40 network may apply to both.

Our work revealed a non-monotonic integration of BCR and CD40R signals in the proliferative responses of B-cells, due to BCR-induced apoptosis and NFκB signaling saturation (Fig. 4). Consistent with previous research which suggested that CD40 signaling alone was sufficient for B-cell affinity maturation (Victora et al, 2010; Shulman et al, 2014), we found that BCR-induced apoptosis prevented BCR stimulation from promoting additional population growth, rendering population expansion primarily CD40-dependent (Figs. 4 and 5). However, BCR signaling provides an important modulatory role in T-dependent selection of B-cells. Chen et al, found BCR signaling to facilitate positive selection by prolonging B-cell survival and by priming B cells to receive synergistic Tfh cell signals (Chen et al, 2023). What we found was consistent with this observation but completed with another part of the story: when costimulated with CD40, BCR signaling modulates

the dose-response curve of CD40 by boosting less-stimulated cells with its pro-proliferative effects yet dampening proliferative responses of more-highly stimulated cells with its anti-survival effects (Figs. 3 and 5). When stimulated alone, BCR regulates B-cell survival in a non-monotonic dose response curve, thereby potentially eliminating cells encoding self-reactive BCRs that elicit strong signals. Indeed, Shih et al, found both more cell division and increased cell death in higher-affinity B1-8$^{hi}$ B-cells compared to lower-affinity B1-8$^{lo}$ cells in post-immunized mice spleen, highlighting again the paradoxical role of BCR stimulation (Shih et al, 2002).

Considering the temporal dynamics of the process, our work delineated a narrow temporal window of opportunity for B-cells to receive CD40 signals following BCR activation. Proper timing (3 h) of the two signals can maximize B-cell survival and proliferative response, while longer delays (8 h) can lead to significant apoptosis and thus reduced population growth (Fig. 6). Consistent with this temporal window, Akkaya et al, also found that BCR signaling activated a metabolic program that imposed a limited time frame (9 h) during which B-cells either receive a second signal (CD40 or CpG) or are eliminated due to mitochondrial dysfunction (Akkaya et al, 2018). In contrast, Tan et al, showed that BCR-induced NR4A nuclear receptors were the key mediators of the restraint on B cell responses to antigen when the cell fails to receive signal 2 within a defined time window, by repressing MYC, and even T-cell chemokines (Tan et al, 2020). Overall, regardless of the exact molecular mediator of the temporal window of opportunity, our model captured the phenotypes described in a large body of literature, and resolved apparently conflicting literature into a more unified systematic model.

Previous work that distinguished the phenotypes between naive and GC B-cells often examined late GC B-cells 10–14 days after immunization (Luo et al, 2018). However, this strategy overlooked the early GC B-cells or the progression of the GC B-cell phenotype necessary for affinity-based selection under different dynamic range. In the early GC phase, where the average antigen-affinity is low, B-cells with mediocre-affinity BCRs need to survive and proliferate. On the other hand, in the late GC phase, where average antigen-affinity is high, the same B-cells with mediocre BCRs would need to avoid proliferating such that B-cells with the highest affinity could be distinguished appropriately. Competition among GC B-cells for antigens and T-cell-help contribute to this flexible dynamic range (Shih et al, 2002). Here, we speculate about a phase-dependent dynamic range in antigen-affinity discrimination, where BCR-induced apoptosis and NFκB signaling saturation together tune the dose-dependent synergy and antagonism between BCR and CD40 signals. The integration of both signals sets a phase-dependent "timer" for B-cell selection. The timer can be further tuned through BCR-induced NFκB signaling, as previous literature suggested that late GC B-cells downregulate their BCR-induced NFκB activation compared to early activated B-cells (Young and Brink, 2021), indicating that remodeling of the BCR signaling network could also contribute to the phase-dependent dynamic range.

Consistent with opposing roles of BCR and CD40 signaling, previous work has suggested that variants that disrupted the signaling of either BCR and CD40 caused an imbalance of positive and negative selection and lead to immunodeficiency or auto-immune diseases. Specifically, Yam-Puc et al, found that enhanced BCR signaling through GC-B-cell-specific SHP-1 mutation led to early GC B-cell death, reducing antibody responses in mice (Yam-Puc et al, 2021). On the other hand, enhanced CD40 signaling through TRAF3 mutations led to autoimmunity and increased risk of B-cell malignancy in humans (Rae et al, 2022), while a lack of CD40 signaling through CD40L mutations led to immunodeficiency in humans (Kroczek et al, 1994).

In summary, our findings may have implications not only for the maturation of high-affinity antibodies but also for the escape of auto-reactive antibodies from negative selection as in autoimmunity. We speculate that the opposing roles of BCR and CD40 signals work together in determining B-cell fates, discriminating highly reactive B-cells as self- versus non-self may amount to a kinetic proofreading mechanism. Specifically, BCR ligand discrimination is due to two signaling steps (through BCR and CD40) that reduce the probability of generating unwanted antibodies. This increased specificity is obtained by introducing cell death, an irreversible step exiting the pathway that happens faster than the next step in the pathway, when the cell receives T-cell-help in the form of CD40 stimulation. Furthermore, this delay between ligand binding and B cell activation consumes free energy due to antigen processing and the activation of cell-death pathway. Understanding this process in greater detail may enable the design of vaccination protocols that maximize B-cell activation and proliferation while ensuring temporal dynamics that selectively induce apoptosis in unwanted B-cell clones, reducing risks of autoimmunity.

# Methods

**Reagents and tools table**

| Reagent/Resource | Reference or Source | Identifier or Catalog Number |
|---|---|---|
| **Experimental models** | | |
| Mouse: C57BL/6 | The Jackson Laboratory | JAX: 000664; RRID: IMSR_JAX: 000664 |
| Mouse: C57BL/6: mTFP1-cRel | The Jackson Laboratory | RRID: IMSR_JAX: 038986 |
| Mouse: C57BL/6: R26-H2B-mCherry | Riken LARGE | Catalog number CDB0239K |
| Mouse: C57BL/6: mVenus-RelA | The Jackson Laboratory | RRID: IMSR_JAX: 038987 |
| **Antibodies** | | |
| Rabbit polyclonal anti-RelA | Santa Cruz Biotechnologies | sc-372; RRID: AB_632037 |
| Rabbit polyclonal anti-cRel | Santa Cruz Biotechnologies | sc-71; RRID: AB_2253705 |
| Rabbit polyclonal anti-p50 | Santa Cruz Biotechnologies | sc-114; RRID: AB_632034 |
| Mouse monoclonal anti-Bcl-xL | Santa Cruz Biotechnologies | sc-8392; RRID: AB_626739 |
| Rabbit polyclonal anti-p84 | Abcam | ab131268 |
| Mouse monoclonal anti-β-tubulin | Sigma-Aldrich | T5201; RRID: AB_609915 |
| HRP Anti-mouse secondary | Cell Signaling Technology | 7076; RRID: AB_330924 |

| Reagent/Resource | Reference or Source | Identifier or Catalog Number |
|---|---|---|
| HRP Anti-rabbit secondary | Cell Signaling Technology | 7074; RRID: AB_2099233 |
| CD40 monoclonal antibody (IC10) | Invitrogen | 16-0401-86; RRID: AB_468940 |
| Goat anti-mouse IgM | Jackson ImmunoResearch | 115-066-020; RRID: AB_2338579 |
| **Chemicals, enzymes, and other reagents** | | |
| Recombinant murine IL-4 | PeproTech | 214-14 |
| Fluorobrite DMEM | Gibco | A1896701 |
| DRAQ7 | ThermoFisher Scientific | D15106 |
| CellTrace™ Far Red Proliferation Kit | ThermoFisher Scientific | C34564 |
| SuperSignal West | ThermoFisher Scientific | 34095, 34580 |
| **Software** | | |
| FlowJo V10.8.1 | FlowJo LLC | N/A |
| FlowMax | (Shokhirev et al, 2015a; Shokhirev and Hoffmann, 2013) | N/A |
| Python v3.7.164-bit base:conda | Anaconda v3.0 | N/A |
| ImageJ2 v2.16.0 | | N/A |
| Julia v1.9.3 | | N/A |
| R v4.2.0 | | N/A |
| **Other** | | |
| Zeiss Axio Observer Wide-field Epifluorescence Microscope | Zeiss | |
| Bio-Rad ChemiDoc XRS Imaging System | Bio-Rad | |
| CytoFLEX Flow Cytometer | Beckman Coulter | |

## Methods and protocols

### Mice

Mice were maintained in a standard environmentally controlled vivarium at the University of California, Los Angeles. Female C57BL/6 mice in each experiment were 8–13 weeks old unless otherwise indicated. Animal work was performed according to guidelines from the University of California, Los Angeles under approved ARC protocols (B-14-110 for breeding and R-14-126 for experiment).

### B cell isolation and culture

Spleens were harvested from 8- to 13-week-old female C57BL/6 mice. Homogenized splenocytes were incubated with α-CD43 magnetic beads for 15 min at 4–8 °C, washed with MACS buffer and passed through an LS column (Miltenyi Biotech). The purity of B cells was assessed at >97% based on B220 staining as described previously (Mitchell et al, 2018). Following isolation, B cells were stimulated with α-CD40 (H: 10 μg/mL, M: 3.3 μg/mL, L: 1 μg/mL), IL-4 (H: 20 ng/mL, M: 6.6 ng/mL, L: 2 ng/mL), and anti-IgM (referred to as α-BCR in the main text, H: 10 μg/mL, L: 1 μg/mL), unless otherwise

specified, and cultured for 4 days in fresh RPMI-based media at 37 °C and 5% $CO_2$. All α-CD40 stimulation conditions mentioned in the results are stimulated with both α-CD40 and IL-4 at corresponding doses.

### Immunoblot

B cells were harvested from culture plates, washed in 1 mL PBS and counted on a CytoFlex flow cytometer (CytoFLEX, Beckman Coulter), prior to preparing lysates for protein content analysis. Due to varying cell sizes and numbers over time as a result of growth and proliferation, an equal number of cells (as opposed to equal protein amounts) per sample was analyzed in each immunoblot. In cases where nuclear fractions were required to be separated, cells were first lysed in CE buffer on ice, followed by vortexing and centrifugation, and the supernatant containing the cytoplasmic fraction was removed. Nuclei were then lysed by 3 repeated freeze-thaw cycles between 37 °C water and dry ice, followed by centrifugation to clear the lysate of nuclear debris, after which the supernatant containing the nuclear fraction was harvested.

For immunoblotting, lysates were run on 4–15% Criterion TGX pre-cast polyacrylamide gels (Bio-Rad), and transferred on to PVDF membranes using wet transfer. The following antibodies were used to identify the proteins of interest: RelA, cRel, Bcl-xL, p84 (loading control for nuclear lysates), and b-tubulin (loading control for cytoplasmic and whole cell lysates). Antibody details are given in the Resources table, and concentrations used were 1:5000 for RelA and cRel, 1:1000 for Bcl-xL, 1:10,000 for p84, and 1:10,000 for b-tubulin. Protein bands were detected using the Bio-Rad ChemiDoc XRS System, with a 10:1 mixture of the SuperSignal West Pico and Femto Maximum Sensitivity Substrates (Thermo Scientific) applied to detect chemiluminescence released by HRP-labeled secondary antibodies.

RelA, cRel, and Bcl-xL bands were quantified by measuring mean gray value using ImageJ2, deducting background value per lane (measured by a box of the same size directly below the target protein band), and normalizing intensities to the 0 h baseline.

### Media and buffer compositions

B cell media: RPMI 1640 (Gibco) supplemented with 100 IU Penicillin, 100 μg/ml Streptomycin, 5 mM L-glutamine, 20 mM HEPES buffer, 1 mM MEM non-essential amino acids, 1 mM Sodium pyruvate, 10% FBS, and 55 μM 2-Mercaptoethanol.

MACS buffer: Phosphate buffered saline (pH 7.4), and 2% bovine serum albumin.

CE Buffer: 50 mM HEPES-KOH pH 7.6, 140 mM NaCl, 1 mM EDTA, 0.5% NP-40, freshly supplemented with EDTA-free protease inhibitors (5 mM DTT, 1 mM PMSF).

NE Buffer: 10 mM Tris-HCl, pH 8.0, 200 mM NaCl, 1 mM EDTA, freshly supplemented with EDTA-free protease inhibitors (5 mM DTT, 1 mM PMSF).

### Measurement of generation-specific B cells by CTFR staining

B cells were stained with Cell Trace Far Red (CTFR) using CellTrace Far Red Cell Proliferation Kit (ThermoFisher Scientific, # C34564) as described by the manufacturer protocol. Briefly, 2 M cells were resuspended in 1 mL RT PBS and incubated with 1 μL CTFR for 25 min at RT with rotation. Cells were washed by centrifugation, resuspension in 1 mL RPMI with 10% FBS, and

incubation for 10 min at RT. The washing steps were repeated 2 more times. CTFR labeled cells were treated with α-CD40, IL-4, and/or anti-IgM for 96 h as described above. The cells were harvested at indicated timepoints and acquired on the CytoFlex flow cytometer (CytoFLEX, Beckman Coulter). The cells were gated based on forward scatter (FSC) and side scatter (SSC) to identify live single cells. Doublets were then excluded from the analysis using FSC area and height. To deconvolve the cells into different generations based on dilution of CTFR, we used the Proliferation Modeling feature on FlowJo V10.8.1. Specifically, generation-0 cells were gated as "Undivided" in the APC-A channel according to the unstimulated control and 24 h samples, and the number of peaks were set based on visual estimation and then further adjusted based on the Peak Ratio and Root Mean Squared outputs to optimize curve fitting. We also note the limitation of the assay in reliably distinguishing division beyond the seventh generation due to limited fluorescence and the nutrient depletion in the medium (Hasbold et al, 1998).

### Microscopy setup

B cells from 10- to 16-week-old cRel$^{T/T}$ RelA$^{V/V}$ H2B$^{C/+}$ mice were plated at about 50,000 cells per well into 24-well TC plates, to maintain optimal density for cell culture and to be compatible with culture conditions in flow cytometry experiments. At each time point, cells were harvested from the original plate, spun down and resuspended in 100 μL Fluorobrite media, and introduced into single wells of a black 96-well plate with #1.5 coverslip glass bottom (Cellvis P96-1.5H-N) to fill up a total of 1 mL Fluorobrite media. Cells were spun down at 500 rcf for 5 min to ensure settling at even density across the glass bottom, and 1 μL DRAQ7 was added as a viability indicator.

The plate was placed on a stage insert in the incubation chamber at 37 °C of a Zeiss Axio Observer wide-field epifluorescence microscope. Immersol 518 F/37 °C was applied to a 63X/NA1.4 Plan Apo oil immersion objective, and contact was made with the glass bottom of the plate. Automated tiling was performed to acquire 49 tiles of 200 μm × 200 μm across each well in a 7 × 7 pattern, separated by a gap of about 40 μm between each tile to avoid overlaps and dual counting of cells. Fluorescence images were collected in 4 channels to measure mTFP1-cRel and mVenus-RelA activity, H2B-mCherry as a nuclear marker, and DRAQ7 for viability, with the following settings:

| Target | Filter | Illumination | Excitation | Dichroic | Emission | Pow | T$_{exp}$ |
|---|---|---|---|---|---|---|---|
| cRel-mTFP1 | Semrock CFP | Colibri 430 nm | 426–443 nm | 452 nm | 460–499 nm | 100% | 700 ms |
| RelA-mVenus | Semrock YFP | Colibri 505 nm | 488–512 nm | 520 nm | 529–556 nm | 100% | 700 ms |
| H2B-mCherry | Semrock mCherry | Colibri 590 nm | 542–582 nm | 593 nm | 604–679 nm | 100% | 300 ms |
| DRAQ7 | Chroma Far Red | XCite HXP-120 lamp | 610–650 nm | 647 nm | 637–697 nm | Max | 100 ms |

### Image processing and fluorescence quantification

Image analysis was semi-automated by using a custom macro in Fiji, which is provided on GitHub (see Data availability). First, a binary mask of dying cells was created using the Moments automatic thresholding method on fluorescence images for DRAQ7. A separate binary mask of the black tiling edges was created and expanded by 25 pixels to filter out cells located at the edge of each tile. Then, whole cells were segmented using the Otsu auto-thresholding method on a fluorescence image calculated from the maximum of cRel-mTFP1 and RelA-mVenus images, followed by morphological processing (fill holes, watershed, and erosion), and particle analysis filtering based on cell size (50–10,000 pixels) and circularity (0.5–1.0). Each whole-cell region of interest (ROI) is then filtered against the binary masks of dying cells and black tiling edges.

Next, nuclear envelopes were also segmented using a user-specified automatic thresholding method (Moments, IsoData, or Intermodes) on Gaussian-filtered fluorescence image for H2B-mCherry, followed by morphological processing, particle analysis filtering based on cell size, circularity, and the cell viability masks. At the end of both segmentations, user is prompted to visually check for segmentation errors and has the option to use Fiji's size scaling and freehand selection tools to manually adjust the ROIs.

Each nuclear ROI was matched to the closest whole-cell ROI, ensuring at least 30% overlap in area. The unmatched nuclear and whole-cell ROIs were deleted. If multiple nuclear ROIs are associated with the same whole-cell ROI (due to having cleaved morphology or being over-segmented in watershed), the nuclei are combined into one using an AND gate. Each whole-cell ROI was then expanded to fully encompass its nucleus using an OR gate, since it occasionally gets too eroded. After this cleanup, cytoplasm ROIs were computed for each whole-cell and nucleus pair, using an XOR gate.

The area of each whole-cell ROI was measured to quantify the distribution of cell sizes in Fig. 3M. All the whole-cell, nuclear, and cytoplasmic ROIs, including a manually drawn background region, were also measured against the raw fluorescence images for mTFP1-cRel, mVenus-RelA, and H2B-mCherry to quantify their background-subtracted median fluorescence intensities as a spatially-smoothed measure of brightness. Because the cells were imaged in different sessions spanning multiple days, we noticed significant fluctuations in fluorescence intensity across timepoints, despite consistent microscopy setup. We thus calculated a normalization factor for each timepoint based on the experimental condition we knew to be the most consistent across time, the cytoplasmic fluorescence of cells stimulated with low α-CD40. The minimum and maximum brightness of the images were adjusted with the normalization factors to ensure visual consistency across timepoints (Fig. EV3). Once the correction factors were visually validated, they were applied to the median fluorescence intensity quantification in Fig. 4F,H.

### Computational modeling of the T-dependent receptor signaling pathway

The mathematical model of T-dependent (TD) B cell stimulation was developed in two parts. First, we expanded the previously published BCR-signaling ODE model (Shinohara et al, 2014; Inoue et al, 2016) by including the BCR receptor antigen binding (Fig. 1A left side), and scaled the parameters to match the units (nM$^{-1}$ h$^{-1}$) in the rest of our model. The Shinohara and Inoue models prescribed a signal function for the CBM complex, a downstream adapter for BCR receptor. We bridged the gap between antigen concentration and CBM signaling with a few additional ODE

equations, and tuned these additional parameters (Table 1) such that the signaling dynamics matched the previous version:

$$\frac{d}{dt}[ANTIGEN] = -\varphi_1 * [ANTIGEN] - \varphi_4[ANTIGEN] * [BCR] \\ * C_{c2m} + \varphi_5 * [ABCR] * C_{c2m}$$

$$\frac{d}{dt}[BCR] = \varphi_2 - \varphi_3 * [BCR] - \varphi_4 * [ANTIGEN] * [BCR] \\ + \varphi_5 * [ABCR]$$

$$\frac{d}{dt}[ABCR] = \varphi_4[ANTIGEN] * [BCR] - \varphi_5 * [ABCR] - \varphi_6 * [ABCR]$$

where [ANTIGEN], [BCR], and [ABCR] are the concentrations of the antigen, BCR, and their complex; $\varphi_i$, $i = 1, 2, 3, \ldots$, are the reaction constants (index are listed in Table 1); $C_{c2m} = 0.01$ is a scaling factor for external ligands like ANTIGEN to convert cellular concentration to media concentration. In this model, [ANTIGEN] is the model input corresponds to experimental stimulation α-BCR. As output of the BCR receptor module, [ABCR] regulates CBM complex activation (Fig. 1A left side).

Next, we abstracted the CD40 model from its known signaling pathway (Elgueta et al, 2009; Akiyama et al, 2012) in a parsimonious way. As mentioned in the discussion, to avoid the complexity of combinatorial biochemical reactions among the TRAF complexes, we used TRAF3 to represent the TRAF2-TRAF3 complex that constitutively inhibits the non-canonical NFκB pathway, and TRAF6 to represent the TRAF1-TRAF2, TRAF3-TRAF5, and TRAF6-TRAF2 complexes that all activate the canonical NFκB pathway.

$$\frac{d}{dt}[CD40L] = -\varphi_{11} * [CD40L] - \varphi_{14}[CD40L] * [CD40R] * C_{c2m} \\ + \varphi_{15} * [CD40LR] * C_{c2m}$$

$$\frac{d}{dt}[CD40R] = \varphi_{12} - \varphi_{13} * [CD40R] - \varphi_{14} * [CD40L] * [CD40R] \\ + \varphi_{15} * [CD40LR]$$

$$\frac{d}{dt}[CD40LR] = \varphi_{14}[CD40L] * [CD40R] - \varphi_{15} * [CD40LR] \\ - \varphi_{16} * [CD40LR]$$

$$\frac{d}{dt}\left[TRAF6_{off}\right] = -\varphi_{17} * [CD40LR] * \left[TRAF6_{off}\right] + \varphi_{18} * \left[TRAF6_{on}\right]$$

$$\frac{d}{dt}\left[TRAF6_{on}\right] = \varphi_{17} * [CD40LR] * \left[TRAF6_{off}\right] - \varphi_{18} * \left[TRAF6_{on}\right]$$

$$\frac{d}{dt}[TRAF3] = \varphi_{19} * [TRAF3] - \varphi_{20} * [TRAF3] - \varphi_{21} * [CD40LR] \\ * [TRAF3]$$

The subsequent kinases that further relay the receptor signal to NFκB signaling are TAK1 (for TRAF6) and NIK (for TRAF3). We used a Hill function for TRAF3-induced degradation of NIK to abstract a more complicated complex formation process:

$$\frac{d}{dt}[NIK] = \varphi_{35} - \varphi_{36} * [NIK] - \varphi_{37} * [NIK] * \frac{[TRAF3]^2}{[TRAF3]^2 + 0.5^2}$$

To generate heterogeneous cell responses, the receptor parameter values were distributed in each virtual B-cell the same way as the NFκB signaling module in previous publication, where synthesis, degradation, association, dissociation rates were drawn from a normal distribution with mean values from the standard parameter set (Table 1) and CV of 11.2% (Mitchell et al, 2018). These ODEs (with 37 parameters and 12 species) were solved using the Tsit5 solver algorithm from the DiffEq.jl package in Julia, with an absolute error tolerance of 1e−5 and relative error tolerance of 1e−3. All simulations were carried out on an Ubuntu server with 64 threads, 2.1 to 3.7 GHz speed, and 384 GB RAM.

### Multiscale modeling coupling signaling network and B cell proliferation

The receptor-extended TD model constructed above was combined with a published MATLAB model of B cell proliferation to create a multiscale model capable of simulating the division and death of a population of individual B-cells upon TD stimulation. The B cell model integrates a biophysically accurate model of canonical NFκB signaling (with about 300 parameters and 61 species) with models of the cell cycle (with 52 parameters and 23 species) and apoptosis (with 117 parameters and 59 species) (Shokhirev et al, 2015a; Mitchell et al, 2018) to create a multiscale model capable of simulating the division and death of a population of individual cells upon T-dependent stimulation. Cleaved PARP (cParp) in the apoptosis model and cadherin-1 (Cdh1) concentration thresholds in the cell cycle model triggered virtual B-cell death and division, respectively. We translated the model from MATLAB into Julia 1.9.3 for faster execution. All reactions and parameters within the NFκB, apoptosis and cell cycle networks were maintained and distributed as described by Mitchell et al (Mitchell et al, 2018), except for 2 parameters in the cell cycle network that were changed to reduce the discrepancy between CD40 and CpG-induced proliferative response (Fig. EV2, see more details in "Local sensitivity analysis to tune CD40-activated cell fates" section of the Methods).

Separate modules of the multi-scale model were employed when simulating for different purposes. For the NFκB dynamics in Fig. 1, only the receptor-NFκB model was used for the simulation, and the cell fate modules (apoptosis and cell cycle) were excluded. For Fig. EV2C–F when we tuned the CD40-activated cell fates, the cell death module was excluded to enable faster turnaround for parameter tuning in the cell cycle module. For Fig. 5B–D, the cell cycle module was excluded to isolate the effects of BCR signaling on cell survival. All the other model simulation used the full multi-scale model. When we reported the population trajectory of NFκB activity in Fig. 1E–H, all 1000 cells contributed to the mean and standard deviation, but in Fig. 4E,G, only cells that are alive at each timepoint contributed to the mean and standard deviation.

All of the code and parameters to run the model simulations and plot the figures is provided on GitHub (see Data availability). A separate parameter table for all the species and reactions is

provided as supplementary materials (Dataset EV2). For each virtual B-cell with its own set of parameters, we ran the model in two phases to first identify the steady state, and then simulate the dynamic time course upon stimulation, with initial states from this steady state. The steady state was solved using Julia's steady state Tsit5 solver with an absolute error tolerance of 1e−5 and relative error tolerance of 1e−3. The simulation time for which the given ODE reach steady state was limited within 800 h.

### Computational modeling of the BCR-induced cell death pathway

Since we found α-BCR stimulation had an NFκB-independent anti-survival effect that overrides its NFκB-dependent pro-survival effect (Fig. 3), we decided to resolve this difference by modifying the multi-scale model. It was reported that ligation of the BCR induces cell death in some B cells (Graves et al, 2004) due to activation of Bcl-2 Interacting Mediator of cell death (Bim) (Gao et al, 2012), caspase-2 or -8 (Chen et al, 1999), mitochondrial dysfunction (Akkaya et al, 2018) or more. Based on these signaling mechanisms that may mediate activation-induced cell death (AICD) in B-cells and the available species in the existing cell death module, we revised the cell death module of the T-dependent multi-scale B-cell model to include a simplified pathway from activated BCR to caspase-8 processing (Fig. 4A):

$$\frac{d}{dt}[PC8] = \left(original\ \frac{d}{dt}[PC8]\right) - \varphi_{C8,AICD} * [PC8] * [ABCR]$$

$$\frac{d}{dt}[C8] = \left(original\ \frac{d}{dt}[C8]\right) + \varphi_{C8,AICD} * [PC8] * [ABCR]$$

where $[PC8]$, $[C8]$, and $[ABCR]$ are the concentrations of the pre-caspase-8, caspase-8, and activated BCR; $original\ \frac{d}{dt}[PC8]$ and $original\ \frac{d}{dt}[C8]$ are the original differential equations for pre-caspase-8 and caspase-8 in Mitchell et al (2018), abbreviated to highlight the revision we made; $\varphi_{C8,AICD}$ was tuned to be 0.00021 according to experimental data of BCR-CD40 costimulation versus CD40-only stimulation conditions (Fig. 4B,C).

Simulations prior to Figs. 4 and 5F–H did not include this BCR-induced cell death pathway. Figures 4B–F, 5A–E,I–K, 6 and 7 were all simulated with the modified caspase-8 equations.

### Model fit evaluation

Root-mean-squared deviation (RMSD) were calculated on the population dynamics between model simulation and experimental results (Figs. 2F, S2B, 3E,F, and 4D) and between two experimental conditions (Fig. 3H,K) in the same manner. Two RMSD scores, one for population expansion index (Fig. 2D), and the other for generational composition (Fig. 2E) between each pair of model and experimental outputs at each experimental timepoint (0, 24, 36, 48, 72, and 96 h) were calculated.

For the RMSD on generational composition:

$$RMSD_{gen} = \sqrt{\sum_{i=1}^{5}\sum_{j=0}^{6}\left(\frac{n_{i,j}}{N_i} - \frac{\hat{n}_{i,j}}{\hat{N}_i}\right)^2}$$

Where $i$ is the $i$-th timepoint of the experimental measurement (i.e., $i = 1$ corresponds to the measurement at 24 h, followed by 36,

48, 72, and 96 h), and $j$ is the generation number, ranging from generation 0 to 6 corresponding to founder cells to cells that have divided 6 times. $n_{i,j}$ thus means the number of live cells in generation $j$ and timepoint $i$ in the experimental data, while $\hat{n}_{i,j}$ is the corresponding live cell number in generation $j$ and timepoint $i$ in model simulation. Additionally, $N_i = \sum_{j=0}^{6} n_{i,j}$ represents the total number of live cells at timepoint $i$ in the experimental data, and $\hat{N}_i = \sum_{j=0}^{6} \hat{n}_{i,j}$ represents the corresponding total live cell number in model simulation. $\frac{n_{i,j}}{N_i}$ and $\frac{\hat{n}_{i,j}}{\hat{N}_i}$ are thus the generation decomposition ratios at each time point for experimental data and simulation data, respectively.

For population expansion, the RMSD is composed of two parts, one normalized to population size at 0 h ($N_0$) and one normalized to the population size at 24 h ($N_1$) to account for unpredictable mechanical cell death (which typically occur within the first few hours) as a form of technical error in experiments. Both RMSD scores are then normalized to the number of timepoints (5 timepoints for 0 h normalization, and 4 timepoints for 24 h normalization) and the maximum population expansion so that different amount of population expansion at different doses are evaluated on the same scale:

$$RMSD_{pop\_exp} = \sqrt{\frac{\sum_{i=1}^{5}\left(\frac{N_t}{N_0} - \frac{\hat{N}_t}{\hat{N}_0}\right)^2}{5 \cdot \max\limits_{i=1,\dots,5} N_i}} + \sqrt{\frac{\sum_{i=2}^{5}\left(\frac{N_t}{N_1} - \frac{\hat{N}_t}{\hat{N}_1}\right)^2}{4 \cdot \max\limits_{i=2,\dots,5} N_i}}$$

### Local sensitivity analysis to tune CD40-activated cell fates

Due to the discrepancy between CD40 and CpG-induced proliferative response, we quantified several key variables in the dye dilution data that determined the population dynamics with FlowMax (Shokhirev and Hoffmann, 2013). After fitting FlowMax model to the experimental data, we quantified the time to first division (Tdiv0), time to later divisions (Tdiv1+), and the fraction of generation 0 cells that respond by dividing (F0) in response to low, medium, and high α-CD40 doses. In all α-CD40 doses, the average Tdiv0 is much later and more dose-specific (68.5 to 76.9 h since stimulation onset for high to low dose of α-CD40) than what the model predicted (36.1 to 40.6 h). On the other hand, the average Tdiv1+ of the α-CD40 experimental data were mostly shorter than predicted by the model (Table 2 first 2 columns, Fig. EV2A), and the proportion of dividers was lower, indicated by a larger amount of cells in generation 0 at 96 h in Fig. EV2A and a smaller F0 quantified by FlowMax than the model predicted (Table 3 first 2 columns).

To improve model fit, we identified locally sensitive parameters in the cell cycle module that contribute to Tdiv0 and Tdiv1+ by calculating the standard deviation in division times when scaling each parameter by 0.2, 0.33, 0.4, 0.5, 0.66, 1.0, 1.5, 2.0, 2.5, 3.0, or 5.0-fold. 2 out of 55 parameters stood out as the best candidates for tuning Tdiv0 and Tdiv1+: retinoblastoma (Rb) decay rate and cyclin B (CycB) synthesis rate, respectively (Fig. EV2C). Rb decay rate was tuned to be 10% of the original value, whereas CycB was tuned to be 1.8-fold the original value to achieve a later and more dose-responsive Tdiv0, shorter Tdiv1+, and smaller divider percentage (Fig. EV2D–F).

A simulation of 300 cells with distributed parameters before and after parameter tuning showed that mean Tdiv0 for dividers increased from 36.14 h to 62.80 h for high dose of α-CD40

**Table 2. Experimental vs model proliferation time before (1) & after (2) tuning.**

| Condition | EXP Tdiv0 | MODEL(1) Tdiv0 | MODEL(2) Tdiv0 | EXP Tdiv1+ | MODEL(1) Tdiv1+ | MODEL(2) Tdiv1+ |
| --- | --- | --- | --- | --- | --- | --- |
| high α-CD40 | 68.5 | 36.1 | 60.7 | 6.1 | 9.0 | 6.2 |
| medium α-CD40 | 68.6 | 35.7 | 66.2 | 7.8 | 9.2 | 6.0 |
| low α-CD40 | 76.9 | 40.6 | 79.4 | 35.2 | 9.7 | 6.3 |

**Table 3. Experimental vs model divider percentage before (1) & after (2) tuning.**

| Condition | EXP F0 | MODEL(1) F0 | MODEL(2) F0 |
| --- | --- | --- | --- |
| high α-CD40 | 46.8% | 58.3% | 46.3% |
| medium α-CD40 | 18.4% | 46.3% | 34.7% |
| low α-CD40 | 4.4% | 24.0% | 21.0% |

stimulation, and from 40.65 h to 74.78 h for low dose, achieving both a later and more dose-responsive Tdiv0, resulting in much more agreement with the FlowMax output based on experimental data (Table 2, left 3 columns). The mean Tdiv1+ for dividers decreases from around 9 h to 6 h for all doses, which is in concordant with high dose of α-CD40, but in less agreement with medium and low doses (Table 3, right 3 columns). Table 3 also showed the percentage of dividers out of all founder cells decreased for all doses, while maintaining α-CD40 dose-responsiveness.

## Data availability

The datasets and computer code produced in this study are available in the following databases: Imaging dataset: BioImage Archive S-BIAD1698. Modeling and image analysis computer scripts: GitHub (https://github.com/helengracehuang/BCR-CD40-integration/).

The source data of this paper are collected in the following database record: biostudies:S-SCDT-10_1038-S44320-025-00124-2.

## Peer review information

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

## Acknowledgements

We thank current and former members of the Hoffmann lab for valuable discussions. We thank Xiaolu Guo, Sohyeon Park, and Aimilia Vareli for critical reading of the manuscript. AH acknowledges funding from sources R01AI132731 and R01AI127867. HVN acknowledges support from the James S McDonnell Foundation Postdoctoral Fellowship and the Damon Runyon Quantitative Biology Fellowship.

## Author contributions

**Helen Huang**: Conceptualization; Software; Formal analysis; Validation; Investigation; Visualization; Methodology; Writing—original draft; Writing—review and editing. **Haripriya Vaidehi Narayanan**: Conceptualization; Resources; Formal analysis; Validation; Methodology; Writing—review and editing. **Mark Yankai Xiang**: Formal analysis; Validation; Writing—review and editing. **Vaibhava Kesarwani**: Validation. **Alexander Hoffmann**: Conceptualization; Supervision; Funding acquisition; Investigation; Writing—review and editing.

Source data underlying figure panels in this paper may have individual authorship assigned. Where available, figure panel/source data authorship is listed in the following database record: biostudies:S-SCDT-10_1038-S44320-025-00124-2.

## Disclosure and competing interests statement

Alexander Hoffmann is a member of the Advisory Editorial Board of Molecular Systems Biology. This has no bearing on the editorial consideration of this article for publication.

# Expanded View Figures

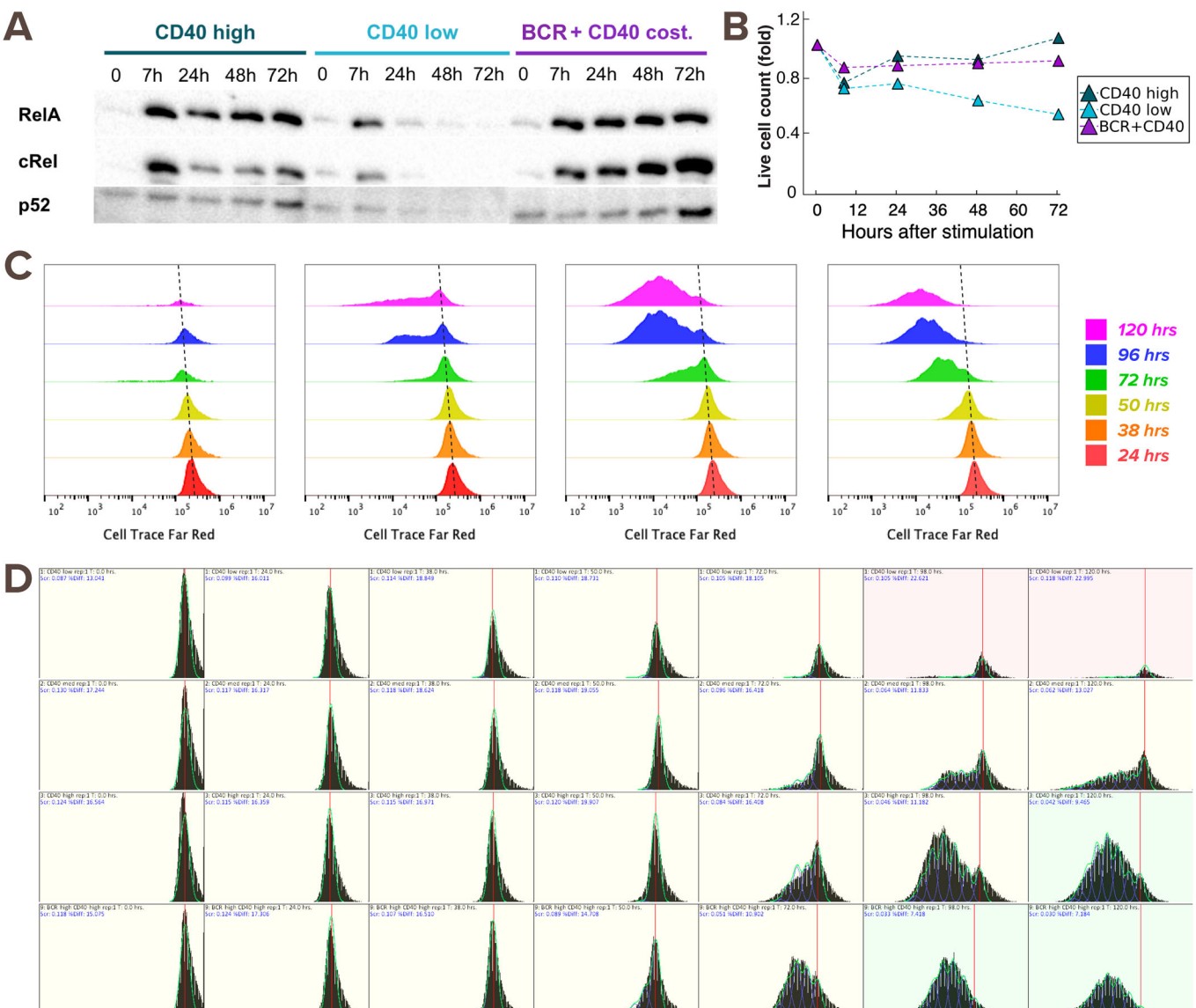

**Figure EV1. Raw experimental data to test the multi-scale B-cell model.**

(A) Immunoblot from experiments with 600 K founder B-cells show nuclear RelA, cRel, and p52 levels at 0, 7, 24, 48, and 72 h after stimulation with low α-CD40 (1 μg/mL), high α-CD40 (10 μg/mL), or costimulation with high α-CD40 and α-BCR (10 μg/mL). (B) Line graph of live B-cell count (fold-change) for each timepoint in (A), to which the samples are adjusted when loading to the gel. The cell count fluctuation is due to cell death, cell division, and technical error when transferring cells. (C) Cell Trace Far Red (CTFR) dye dilution fluorescence histogram for B-cells stimulated with (from left to right) low (1 μg/mL), medium (3.3 μg/mL), and high (10 μg/mL) dose of α-CD40 and costimulation of high α-CD40 and α-BCR (10 μg/mL). There is a baseline shift in CTFR fluorescence by about 2-fold from 24 h to 120 h (dotted line), which we adjusted when deconvolving the cells into each generation. (D) Deconvolution of the time courses in (C) into each generation, where the red line indicates the center of the undivided population of cells, the blue lines indicate individual proliferation peaks, and the green line represents the model sum.

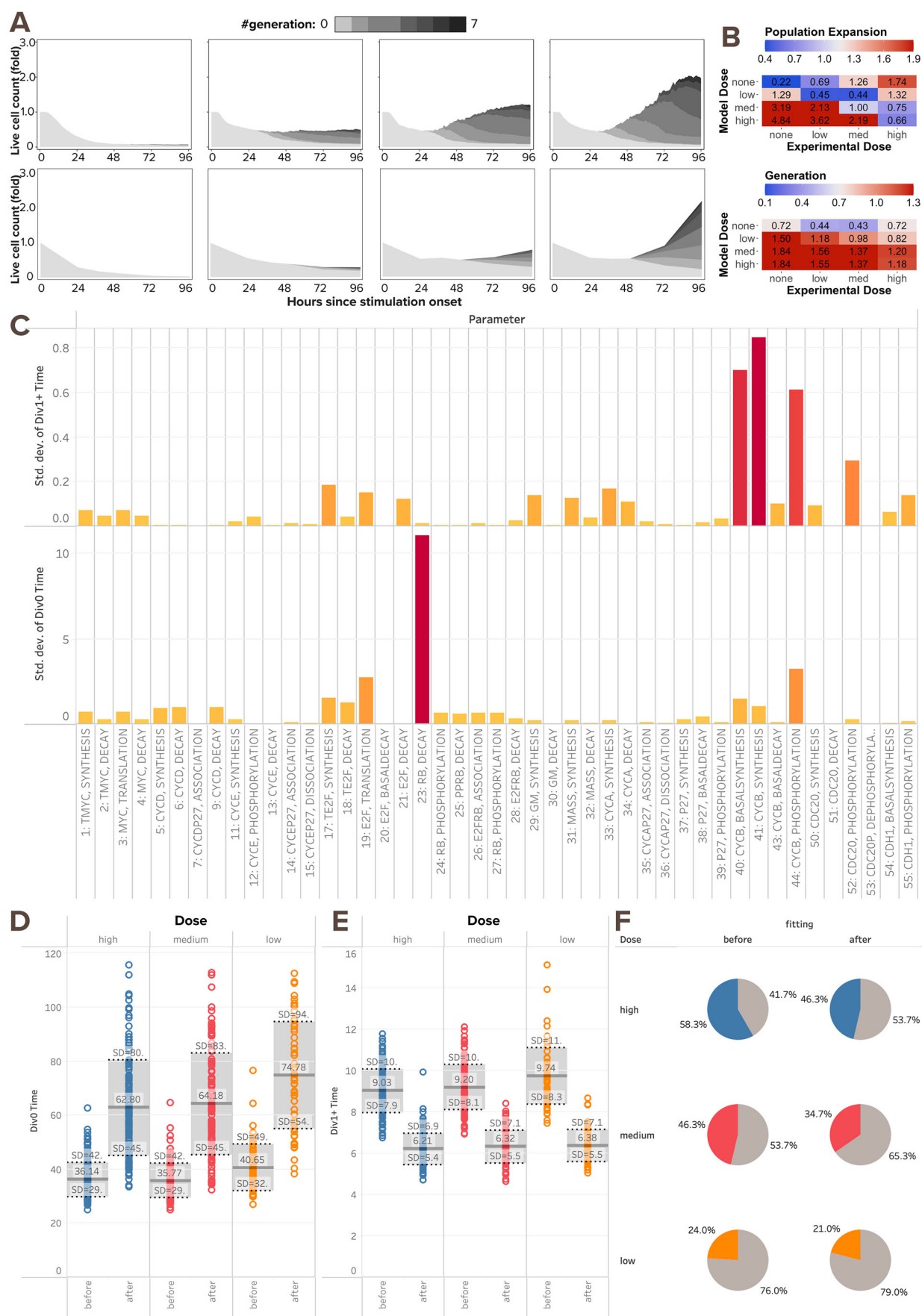

◀

**Figure EV2. Multi-scale model needs tuning to recapitulate B-cell population dynamics in response to CD40 stimulation.**

(A) Stacked area plots from model simulations of 1000 virtual B-cells (top) and matching experiments with 19196 founder B-cells (bottom) show their population dynamics in response to stimulation with (from left to right) no (0 nM and 0 µg/mL), low (6 nM and 1 µg/mL), medium (12 nM and 3.3 µg/mL), and high (30 nM and 10 µg/mL) dose of α-CD40. Each subsequent generation of proliferating cells is indicated with a darker gray. (B) Heatmap shows RMSD of relative population size expansion (top) and generation composition (bottom) in matching (diagonal) or mismatching (off-diagonal) model-and-experiment pairs. Some model doses (medium and low) are more deviated from their matching than mismatching experimental doses (high and medium, respectively), indicating a subpar fit. (C) Bar graph from local sensitivity analysis of parameters in the cell cycle module shows their standard deviations in time to first division ($T_{div0}$) and time to later divisions ($T_{div1+}$). Local sensitivity analysis is achieved by repetitive simulations that independently scaling each parameter in the cell cycle module by 0.2, 0.33, 0.4, 0.5, 0.66, 1.0, 1.5, 2.0, 2.5, 3.0, or 5.0-fold. 2 out of 55 parameters stand out as the best candidates for tuning $T_{div0}$ and $T_{div1+}$: retinoblastoma (Rb) decay rate and cyclin B (CycB) synthesis rate, respectively. These parameters were tuned in order to achieve a later and more dose-responsive $T_{div0}$, shorter $T_{div1+}$, and smaller divider percentage. (D) Box plots from model simulations of 300 virtual B-cells show the mean $T_{div0}$ increases for all doses after parameter tuning. (E) Box plots from model simulations of 300 virtual B-cells show the mean $T_{div1+}$ decreases for all doses after parameter tuning. (F) Pie charts from model simulations of 300 virtual B-cells show the percentage of dividing cells (colored slices) out of all founder cells decreases for all doses, while maintaining CD40 dose-responsiveness. Gray slices are the non-dividing founder cells that either die or survive without division.

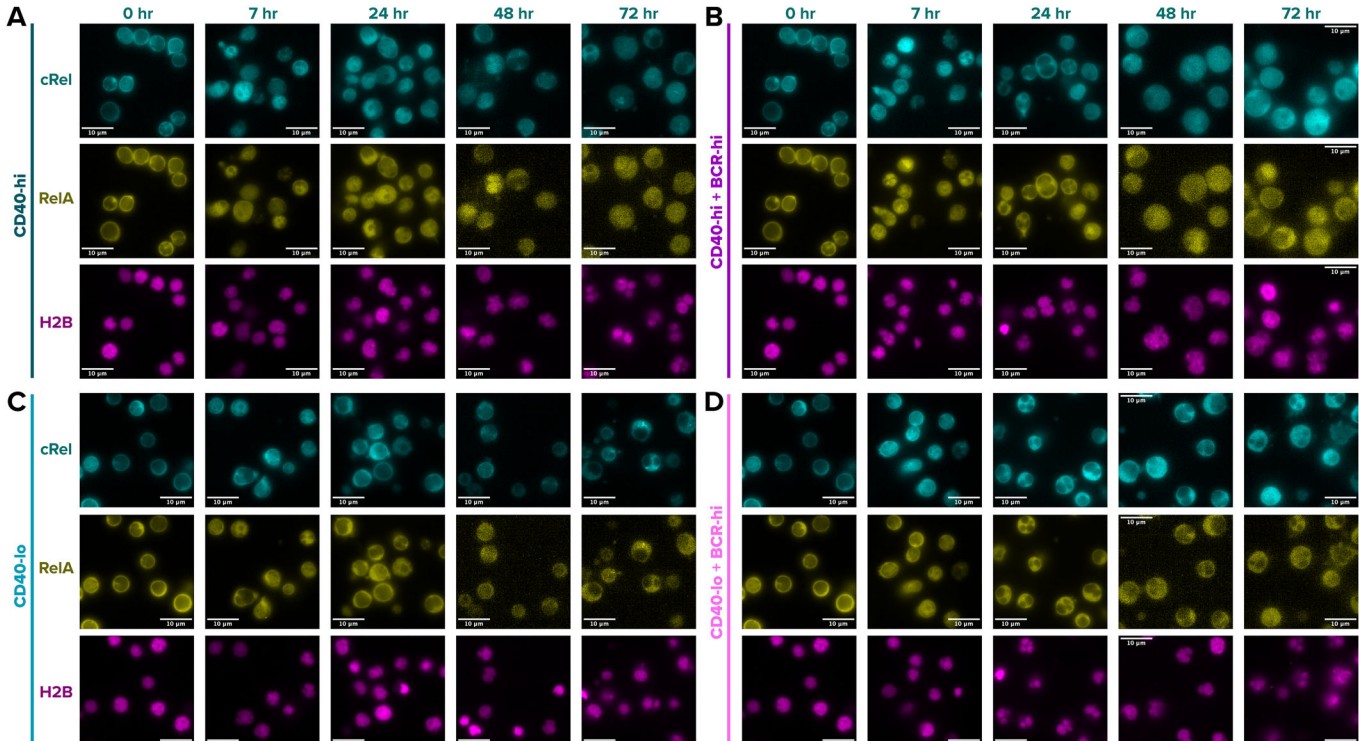

**Figure EV3.   Batch-normalized, background subtracted multi-channel microscopy captures single-cell NFκB dynamics.**

(A–D) Multi-channel fluorescence microscopy images of live triple-reporter B-cells at 63X/NA1.4 under oil immersion. Panels from top to bottom show mTFP1-cRel (blue) and mVenus-RelA (yellow) cellular localization, with H2B-mCherry (pink) as a marker distinguishing nuclear from cytoplasmic compartment, all compared from left to right at 0 h baseline, 7 h, 24 h, 48 h, and 72 h post-stimulation with (A) high α-CD40 (10 μg/mL), (B) high α-BCR (10 μg/mL) and high α-CD40 (10 μg/mL), (C) low α-CD40 (1 μg/mL), and (D) high α-BCR (10 μg/mL) and low α-CD40 (1 μg/mL).

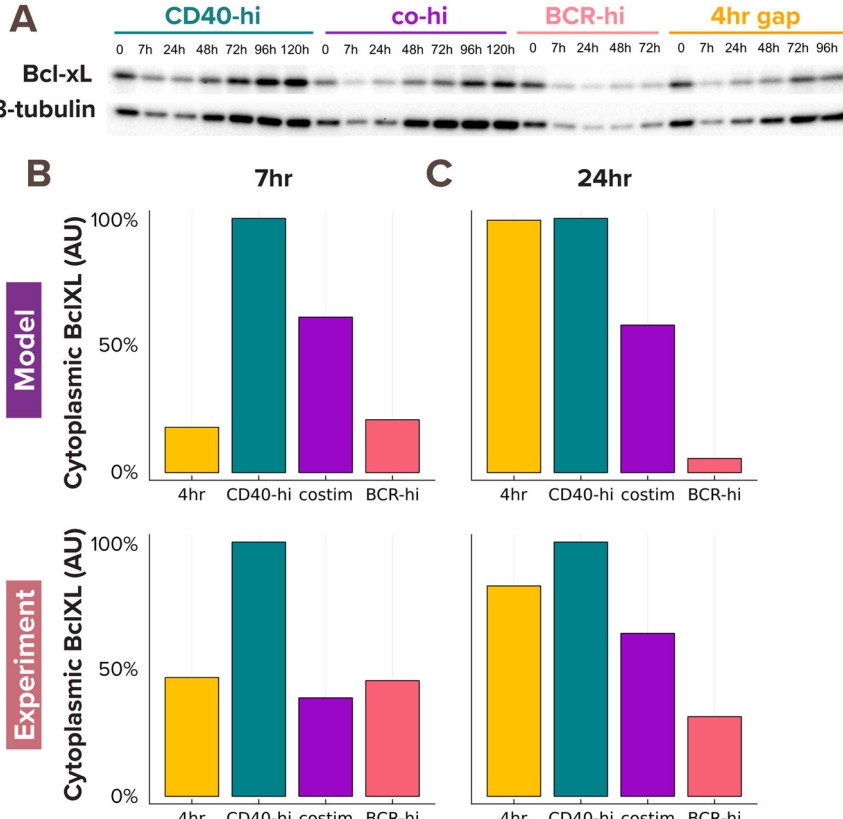

**Figure EV4. Model-simulated cytoplasmic BclXL level recapitulates experimental results.**

(A) Immunoblot from experiments with 600 K founder B-cells show cytoplasmic Bcl-xL and β-tubulin levels in response to stimulation with (from left to right) high (10 μg/mL) dose of α-CD40, high α-CD40 and high α-BCR, high α-BCR, and sequential stimulation of high α-BCR and high α-BCR with a 4 h delay. (B, C) Bar graphs from model simulations (top) and experiments (bottom) show consistent max-normalized quantification of cytoplasmic Bcl-xL level at (B) 7 h and (C) 24 h.

