## [Peer Review File · Molecular Systems Biology]

Synergy and antagonism in the integration of BCR and CD40 signals that control B-cell proliferation

Helen Huang, Haripriya Vaidehi Narayanan, Mark Xiang, Vaibhava Kesarwani, and Alexander Hoffmann

Corresponding author(s): Alexander Hoffmann (ahoffmann@ucla.edu)

Review Timeline:

Transfer from Review Commons:	27th Oct 24
Editorial Decision:	30th Oct 24
Revision Received:	21st Mar 25
Editorial Decision:	29th Apr 25
Revision Received:	16th May 25
Accepted:	19th May 25

The Review Commons logo, with "Review" in a large, blue, italicized serif font and "COMMONS" in a smaller, blue, all-caps sans-serif font below it.

Editor: Jingyi Hou

Transaction Report: This manuscript was transferred to Molecular Systems Biology following peer review at Review Commons.

Review #1

1. Evidence, reproducibility and clarity:

Evidence, reproducibility and clarity (Required)

****Summary:****

This manuscript uses a mathematical model to explore integration of B cell receptor (BCR) and CD40 stimulation of B cells and how it affects the signaling and proliferation response. The authors found that to accurately model B cell responses to co-stimulation required connecting CD40-induced signaling to a proliferation module and BCR-induced signaling to an activation induced cell death module. The authors showed that BCR stimulation synergizes with CD40 signaling at low doses to amplify proliferation but at high doses, the BCR induces apoptosis in founder cells that antagonizes proliferation induced by CD40. The interaction between these modules is time-dependent, which the author suggest can tune the positive and negative selection of B cells. The conclusions are supported by a mathematical model that is well described and with code available. Many key predictions of the model are tested with experimental data.

****Major comments:****

- Fig. 4: A major conclusion of this figure is that "NF- κ B signaling saturation explains the dose-dependent interaction in costimulation" however Fig. 4F does not have experimental validation showing that RelA and cRel activity stimulated by low-dose α -CD40 are in fact amplified by α -IgM at high dose as the model predicts. It seems important to include this data point, especially since the high dose experimental points are rather noisy.
- Regarding the immunoblot experiments, there are no error bars included for any of the immunoblot data. Were these done with replicates?
- Fig. 6: Another key conclusion is that noise in BCR-induced Bcl-xL expression, as well as NF- κ B activity, determines the balance of apoptosis and proliferation in cell populations. The simulations in B and C predict levels of heterogeneity in key signaling molecules. Testing this experimentally using flow cytometry for some of the molecules (e.g., I κ B α as a proxy for RelA activity and caspase 8) would strengthen these predictions.

****Minor comments:****

- Line 30: "co-stimulation with α -CD40 and α -BCR". The authors refer to α -IgM rather than α -BCR for the rest of the paper and so it's confusing to write it differently here.

- Fig. 6C: it looks like some smoothing was done on these experimental curves but I couldn't find where the method was described.

2. Significance:

Significance (Required)

A strength of this study is the use of a mathematical model to consider complex questions of timing and signal integration that are difficult to thoroughly explore experimentally due to time/cost and difficulty of interpreting of the results. The authors show that the model provides biologically relevant insights and include some experimental validation, which is also a strength. A limitation is that some of the more complex predictions are difficult to experimentally validate. However, it would improve the study and likely be more convincing to non-mathematical immunologists to expand some of this validation.

Advance: In systems biology, this study is a novel example of integrating mathematical models at different time scales, and understanding mechanisms underlying positive and negative selection is an important basic research area. I do not study B cell biology, and so I'm less familiar with how it fits into the current literature in that field.

Audience: The audience for this paper would be both systems biologists integrating modeling and experiment to study key signaling pathways, as well as B cell immunologists interested in this specific application area.

3. How much time do you estimate the authors will need to complete the suggested revisions:

Estimated time to Complete Revisions (Required)

(Decision Recommendation)

Between 1 and 3 months

No

Review #2

1. Evidence, reproducibility and clarity:

Evidence, reproducibility and clarity (Required)

****Summary:****

The study investigates how BCR and CD40 signaling influence B-cell fate during the humoral immune response. Using a differential-equations-based model of NF- κ B activation, the model accurately predicts NF- κ B dynamics following co-stimulation, but initially overestimates B-cell population expansion compared to stimulation experiments. This discrepancy is attributed to BCR-induced caspase activity leading to apoptosis unless counteracted by timely NF- κ B-driven survival signals. Sequential co-stimulation experiments confirm that the temporal dynamics of BCR and CD40 signaling determine the balance between positive and negative B-cell selection during T-cell activation.

****Major:****

- The authors need to more clearly distinguish between naive B cell activation by antigens and T-cells, and the affinity maturation process of centrocytes in the germinal center (GC), especially in the introduction and discussion. Although the study focused solely on naive B-cell activation, the findings were extrapolated to centrocytes affinity maturation in several instances (line 504, 561). While the term "founder cells" was used, the distinction should be made clearer, particularly in the discussion.
- Given the similarities between the NF- κ B pathway induced by antigens in founder cells and by follicular dendritic cells (FDCs) in centrocytes, the authors could broaden the impact of their findings by exploring or discussing how their findings might also apply to centrocyte affinity maturation. However, before delving into this, it is crucial to clearly explain the differences between FDC peptide uptake and antigen activation in the context of the NF- κ B pathway.

****Minor:****

- Figure 1: The title, "Multi-scale model recapitulates B-cell NF κ B dynamics in response to T-dependent stimulation," is misleading, as the model does not accurately capture the dynamics at later time points. Additionally, it cannot be considered truly multiscale since it omits key processes such as B-cell division and death. The title should be revised to better reflect the figure.
- Figure 1A: Since follicular dendritic cells (FDCs) are not involved in the activation of founder cells but only in the selection of centrocytes, they should be removed from the figure.
- Figure 1A: The description, "Schematic of BCR-CD40 receptor model to recapitulate T-dependent activation of B-cells," needs more detail. What do the numbers represent? What do the circles with a red 'S' signify? These elements should be clarified in the figure's explanation.
- Figure 1C,D: What do A:50, A:52, C:50, and C:52 represent? Please provide an explanation for these notations in the Figure's caption.
- Figures 3 and 4: The "hi" and "low" labels should be reversed in one of the figures to ensure consistency between the two.
- Figure 4: A more detailed schematic of caspase-8 (including PC8, C8, and ABCR) would be helpful, as this is the authors' main contribution to explain the experimental data.
- Table 1: The table currently only covers the model described in section one. It would be beneficial to add information about the complete multiscale model, highlighting the number of parameters from other pathways and the newly introduced caspase-8 pathway. Either expand Table 1 or create a new table to include this additional information.

2. Significance:

Significance (Required)

This work shed light into the complex mechanisms of founder B cell activation through antigen and T-cell co-stimulation, providing a systematic study that includes experimental validation to refine and enhance existing gene regulatory networks (GRNs). This work is particularly significant as it offers a detailed quantitative assessment of a previously unknown co-stimulation in BCR activation, supported by experimental evidence and convincing quantitative measures. However, the study's use of a highly complex system with several hundreds of parameters raises concerns about the feasibility of integrating this GRN into a full scale germinal center mechanistic model [such as doi.org/10.1038/s41540-023-00271-y, <https://doi.org/10.3390/cells9061448>]. Simplifying the model by focusing on a few critical interactions and parameters could be beneficial to the community, as it has been previously done for example with Memory vs. Plasma blast B-cell differentiation (doi.org/10.1073/pnas.111301910), which utilized only a dozen of reactions. Moreover, the

study's focus on Naive B cell activation, without addressing the role of centrocytes during the germinal center reaction, limits its scope to B cell activation rather than the broader process of B cell affinity maturation. Despite these limitations, the discovery of a new potential mechanism represents a significant advancement in the field. The paper is likely to appeal to a specialized audience, particularly those familiar with mechanistic models and the pathways involved in the B-cell cycle, as well as T-cell and antigen interactions.

3. How much time do you estimate the authors will need to complete the suggested revisions:

Estimated time to Complete Revisions (Required)

(Decision Recommendation)

Between 1 and 3 months

Yes

Revision Plan

Manuscript number: RC-2024-02622

Corresponding author(s): Alexander, Hoffmann

[The “revision plan” should delineate the revisions that authors intend to carry out in response to the points raised by the referees. It also provides the authors with the opportunity to explain their view of the paper and of the referee reports.]

The document is important for the editors of affiliate journals when they make a first decision on the transferred manuscript. It will also be useful to readers of the reprint and help them to obtain a balanced view of the paper.

*If you wish to submit a full revision, please use our "Full Revision" template. **It is important to use the appropriate template to clearly inform the editors of your intentions.**]*

1. General Statements [optional]

This section is optional. Insert here any general statements you wish to make about the goal of the study or about the reviews.

We included point-by-point response to the reviewer evaluation of the significance of the paper with clarification of the goal of the study.

Response to reviewer 1 (reviewer 1 comment in black, our statement in green):

We thank the reviewer for their careful reading of the paper and their appreciation of the computational model and the experimental testing of predictions.

A strength of this study is the use of a mathematical model to consider complex questions of timing and signal integration that are difficult to thoroughly explore experimentally due to time/cost and difficulty of interpreting of the results. The authors show that the model provides biologically relevant insights and include some experimental validation, which is also a strength. A limitation is that some of the more complex predictions are difficult to experimentally validate. However, it would improve the study and likely be more convincing to non-mathematical immunologists to expand some of this validation.

We thank the reviewer for pointing out the strengths of the study. However, we would like to gently push back on the criticism: a math model has the same function in research as an experimental model. To enable discoveries, while being cognizant of its limitations.

Experimental models (mice, in vitro cell culture) can provide some measurements, but are sometimes limited in drilling down into the molecular mechanism, or unable to characterize emergent properties. This is where math models can excel and complement experimental models. This is the role our math model plays here: we connect detailed molecular mechanism with some emergent properties like “window of opportunity” in sequential stimulation.

Of course, we do understand that the math model must recapitulate experimental measurements so that we can trust the insights from the model. We have emphasized the many experimental results already contained in the paper, and we will undertake additional

Revision Plan

experimentation as outlined above.

Advance: In systems biology, this study is a novel example of integrating mathematical models at different time scales, and understanding mechanisms underlying positive and negative selection is an important basic research area. I do not study B cell biology, and so I'm less familiar with how it fits into the current literature in that field.

Audience: The audience for this paper would be both systems biologists integrating modeling and experiment to study key signaling pathways, as well as B cell immunologists interested in this specific application area.

We appreciate the assessment. We agree that the work will be of interest to both systems biologists and B-cell immunologists. Computational models are of increasing importance and relevance to biological studies and the current work documents a significant advance. We will add additional experimental work, as described, to ensure that experimentally trained biologists have confidence in the insights generated by the computational model which represents an integration of prior knowledge of experimental biology studies.

Response to reviewer 2 (reviewer 2 comment in black, our statement in green):

This work shed light into the complex mechanism of founder B cell activation through antigen and T-cell co-stimulation, providing a systematic study that includes experimental validation to refine and enhance existing gene regulatory networks (GRNs). This work is particularly significant as it offers a detailed quantitative assessment of a previously unknown co-stimulation in BCR activation, supported by experimental evidence and convincing quantitative measures. We thank the reviewer for appreciating the significance of providing a detailed model of known molecular mechanisms to explore B-cell responses to stimulation, and the experimental support for this effort.

However, the study's use of a highly complex system with several hundreds of parameters raises concerns about the feasibility of integrating this GRN into a full scale germinal center mechanistic model [such as doi.org/10.1038/s41540-023-00271-y, <https://doi.org/10.3390/cells9061448>]. Simplifying the model by focusing on a few critical interactions and parameters could be beneficial to the community, as it has been previously done for example with Memory vs. Plasma blast B-cell differentiation (doi.org/10.1073/pnas.111301910), which utilized only a dozen of reactions. Moreover, the study's focus on Naive B cell activation, without addressing the role of centrocytes during the germinal center reaction, limits its scope to B cell activation rather than the broader process of B cell affinity maturation.

We thank the review for raising concerns of using a highly complex model with several hundreds of parameters. We'd like to address two aspects to this:

- 1) It is important to distinguish model constructions that attempt to produce highly complex model *de novo*, versus research in which complex models are the result of a decades long process of iterative construction and experimental validation. That is the case in the present model. For example, the NFκB signaling model has been parameterized extensively with biochemical and biophysical studies between 2002 and 2012 and recently was shown to capture the NFκB system regulation in a variety of cell types and stimulus conditions (Mitchell et al 2023). Similar effector modules of apoptosis and cell cycle have undergone rigorous analysis

in the groups of Peter Sorger and John Tyson, and our own lab has vetted these for B-cells in four prior publications (MSB 2015, PNAS 2018, Immunity 2019, PNAS 2024). Therefore, the present model represents a (growing) knowledge base of the molecular mechanisms that govern B-cell responses to stimulation.

2) We agree with the reviewer that in order to simulate B-cell responses at the organ level or organism level, the present level of detail may be too cumbersome, and may be computationally too expensive. We thank the reviewer for pointing out some papers that have presented elegant models with fewer parameters that address the organ level outcomes of the germinal center reaction in terms of affinity maturation. However, these models do not contain the molecular mechanisms that may be affected by genetic variants or may potentially be targeted pharmacologically. We agree with the reviewer that the detailed model we present would need to be carefully pared down to preserve molecular mechanisms of interest while rendering the simulations less computationally expensive.

Despite these limitations, the discovery of a new potential mechanism represents a significant advancement in the field. The paper is likely to appeal to a specialized audience, particularly those familiar with mechanistic models and the pathways involved in the B-cell cycle, as well as T-cell and antigen interactions.

We are pleased that the reviewer appreciated that we report a significant advance for the field, as we reveal a key mechanism that governs the stringency of B-cell selection. It shows how negative and positive selection are intertwined, with BCR mediating negative selection, while enabling positive selection that comes from T-cells via CD40.

2. Description of the planned revisions

Insert here a point-by-point reply that explains what revisions, additional experimentations and analyses are planned to address the points raised by the referees.

Reviewer 1 Major comments (reviewer 1 comment in black, our planned revision in red):

- Fig. 4: A major conclusion of this figure is that "NF- κ B signaling saturation explains the dose-dependent interaction in costimulation" however Fig. 4F does not have experimental validation showing that RelA and cRel activity stimulated by low-dose α -CD40 are in fact amplified by α -IgM at high dose as the model predicts. It seems important to include this data point, especially since the high dose experimental points are rather noisy.
- Regarding the immunoblot experiments, there are no error bars included for any of the immunoblot data. Were these done with replicates?

We agree with the reviewer that additional datasets would be helpful in further informing or testing the model. Rather than undertaking more immunoblots that represent poorly quantifiable population averages, we plan to take a single-cell resolution experimental approach using splenic B-cells from our recently generated RelA and cRel fluorescent reporter mice (Vaidehi Narayanan et al. *PNAS* 2024). We also plan to complement static measurements with live cell microscopy to measure the trajectories of nuclear RelA and cRel activity. That will allow us to obtain distributions of RelA and cRel activity as well as their time-dependent evolution. This will be a significant improvement over the immunoblots in both single-cell and temporal resolution.

Revision Plan

• Fig. 7: Another key conclusion is that noise in BCR-induced Bcl-xL expression, as well as NF- κ B activity, determines the balance of apoptosis and proliferation in cell populations. The simulations in B and C predict levels of heterogeneity in key signaling molecules. Testing this experimentally using flow cytometry for some of the molecules (e.g., I κ B α as a proxy for RelA activity and caspase 8) would strengthen these predictions.

This is a good suggestion. We plan to do intracellular staining for the molecular species of interest, such as Bcl-xL, caspase-8. At each timepoint after stimulation, we will collect the sample, fix and permeabilize the cells, then block and stain for BclXL, caspase-8 with the corresponding antibodies. For RelA and cRel, we plan the live cell imaging work with the fluorescent protein reporters described above. Alternatively, we will do immunostaining in conjunction with DAPI and plate the cells for imaging on the microscope, followed by computational image segmentation of nuclear activity. We will then compare the degrees of heterogeneity between our model simulations and experimental measurements from flow cytometry and imaging.

Reviewer 2 Major comments (reviewer 2 comment in black, our planned revision in red):

- The authors need to more clearly distinguish between naive B cell activation by antigens and T-cells, and the affinity maturation process of centrocytes in the germinal center (GC), especially in the introduction and discussion. Although the study focused solely on naive B-cell activation, the findings were extrapolated to centrocytes affinity maturation in several instances (line 504, 561). While the term "founder cells" was used, the distinction should be made clearer, particularly in the discussion.

We apologize for the confusion between naïve B cells and GC B cells in the introduction and discussion. The reviewer's comments provide a helpful roadmap for the fix. We agree with the reviewer that the findings focus on naïve founder cells, and in the revised paper will clarify that the findings pertain to the initial negative and positive selection event and how that affects antibody responses, but that subsequent steps involving GC centrocytes may be subject to different control mechanisms. Affinity maturation is therefore affected primarily by the initial negative vs positive selection detailed in this paper.

- Given the similarities between the NF- κ B pathway induced by antigens in founder cells and by follicular dendritic cells (FDCs) in centrocytes, the authors could broaden the impact of their findings by exploring or discussing how their findings might also apply to centrocyte affinity maturation. However, before delving into this, it is crucial to clearly explain the differences between FDC peptide uptake and antigen activation in the context of the NF- κ B pathway.

We agree with the reviewer that it's crucial to explain the differences between pre-GC B-cells vs GC light zone centrocytes, and the role of negative selection in each case. We plan to include clear description of this difference in the discussion: Pre-GC B-cells are naive B-cells that recognize soluble or membrane-bound antigens through their BCR, typically in lymphoid tissue before the formation of germinal center. This BCR engagement directly activates the NF κ B pathway through a series of adapter proteins. On the other hand, centrocytes in the GC sample and uptake antigens that are presented in immune complexes on the surface of follicular dendritic cells (FDCs) through their BCR. It has been shown that centrocytes downregulate their

Revision Plan

BCR-induced NF κ B activation compared to early activated B-cells (Young and Brink, Immunity 2021).

Reviewer 2 Minor comments:

- Table 1: The table currently only covers the model described in section one. It would be beneficial to add information about the complete multiscale model, highlighting the number of parameters from other pathways and the newly introduced caspase-8 pathway. Either expand Table 1 or create a new table to include this additional information.

We agree with the reviewer about the importance of a master parameter table for the complete multiscale model, and apologize for the oversight. We will present such a table in the supplement.

3. Description of the revisions that have already been incorporated in the transferred manuscript

Please insert a point-by-point reply describing the revisions that were already carried out and included in the transferred manuscript. If no revisions have been carried out yet, please leave this section empty.

Reviewer 1 Minor comments (reviewer 1 comments in black, our revisions in purple):

• Line 30: "co-stimulation with a-CD40 and a-BCR". The authors refer to a-IgM rather than a-BCR for the rest of the paper and so it's confusing to write it differently here.

We thank the reviewer for pointing this out. The nomenclature was indeed confusing when we use a-BCR and a-IgM interchangeably. Because we focus on the signaling function of the BCR, we have decided to change all the mentions of "α-IgM" to "α-BCR" in the paper. We added that "α-BCR" referred to "α-IgM" in the Materials and Methods (B cell isolation and culture).

• Fig. 6C: it looks like some smoothing was done on these experimental curves but I couldn't find where the method was described.

We apologize for missing the description for this smoothed line. We have added description of the smoothed line to the figure caption.

Reviewer 2 Minor comments (reviewer 2 comments in black, our revisions in purple):

- Figure 1: The title, "Multi-scale model recapitulates B-cell NF κ B dynamics in response to T-dependent stimulation," is misleading, as the model does not accurately capture the dynamics at later time points. Additionally, it cannot be considered truly multiscale since it omits key processes such as B-cell division and death. The title should be revised to better reflect the figure.

We thank the reviewer for pointing out this important point. The model presented in Figure 1 is indeed not multi-scale – only the models in later figures are. We have revised the title to more accurately reflect this fact and the results.

- Figure 1A: Since follicular dendritic cells (FDCs) are not involved in the activation of founder cells but only in the selection of centrocytes, they should be removed from the figure.

We thank the reviewer for pointing out this discrepancy in the figure and the biology we are

Revision Plan

presenting so we have removed the FDCs and T cells from the figure to avoid confusion.

- Figure 1A: The description, "Schematic of BCR-CD40 receptor model to recapitulate T-dependent activation of B-cells," needs more detail. What do the numbers represent? What do the circles with a red 'S' signify? These elements should be clarified in the figure's explanation. We apologize for the lack of detail. In the revision we have added explanations for each of the elements in the caption and revised the figure to include an expanded legend.

- Figure 1C,D: What do A:50, A:52, C:50, and C:52 represent? Please provide an explanation for these notations in the Figure's caption.

We apologize for not explaining the short hand: these are the NF κ B dimers: RelA:p50, cRel:p50 etc. In the revision we have now added explanations in the figure caption.

- Figures 3 and 4: The "hi" and "low" labels should be reversed in one of the figures to ensure consistency between the two.

We thank the reviewer for pointing this out. To maintain visual consistency between figure 3 and 4, we have switched the panels in figure 4.

- Figure 4: A more detailed schematic of caspase-8 (including PC8, C8, and ABCR) would be helpful, as this is the authors' main contribution to explain the experimental data.

We agree with the reviewer that this is an important figure. We have now added a more detailed schematic of caspase-8 processing that involved PC8, C8, and ABCR to Fig. 4A. The kinetic constant will be added to the supplement parameter table mentioned in the next comment.

4. Description of analyses that authors prefer not to carry out

Please include a point-by-point response explaining why some of the requested data or additional analyses might not be necessary or cannot be provided within the scope of a revision. This can be due to time or resource limitations or in case of disagreement about the necessity of such additional data given the scope of the study. Please leave empty if not applicable.

30th Oct 2024

Manuscript Number: MSB-2024-12724-T

Title: Synergy and antagonism in the integration of BCR and CD40 signals that control B-cell proliferation

Author: Helen Huang

Haripriya Vaidehi Narayanan

Alexander Hoffmann

Dear Alex,

Thank you for the submission of your manuscript to Molecular Systems Biology. I have now had a chance to carefully read your manuscript and revision plan. We think the study seems interesting, and we would like to invite a major revision of your manuscript.

All issues raised by the reviewers need to be satisfactorily addressed. Your revision plan appears reasonable, and we would encourage you to proceed with the revisions as outlined.

As you may already know, our editorial policy allows in principle a single round of major revision, and it is therefore essential to respond to the reviewers' comments that are as complete as possible.

On a more editorial level, we would ask you to address the following issues:

- Please provide a .docx formatted version of the manuscript text (including legends for main figures, EV figures and tables). Please make sure that the changes are highlighted to be clearly visible.

- Please provide individual production quality figure files as .eps, .tif, .jpg (one file per figure).

- Please provide a .docx formatted letter INCLUDING the reviewers' reports and your detailed point-by-point responses to their comments. As part of the EMBO Press transparent editorial process, the point-by-point response is part of the Review Process File (RPF), which will be published alongside your paper.

- Please note that all corresponding authors are required to supply an ORCID ID for their name upon submission of a revised manuscript.

- We replaced Supplementary Information with Expanded View (EV) Figures and Tables that are collapsible/expandable online (see examples in <http://msb.embopress.org/content/11/6/812>). A maximum of 5 EV Figures can be typeset. EV Figures should be cited as 'Figure EV1, Figure EV2' etc... in the text and their respective legends should be included in the main text after the legends of regular figures.

Additional Tables/Datasets should be labeled and referred to as Table EV1, Dataset EV1, etc. Legends have to be provided in a separate tab in case of .xls files. Alternatively, the legend can be supplied as a separate text file (README) and zipped together with the Table/Dataset file.

For the figures and tables that you do NOT wish to display as Expanded View figures, they should be bundled together with their legends in a single PDF file called *Appendix*, which should start with a short Table of Content. Each legend should be below the corresponding Figure/Table in the Appendix. Appendix figures and tables should be referred to in the main text as: "Appendix Figure S1, Appendix Figure S2, Appendix Table S1" etc. See detailed instructions regarding expanded view here: <https://www.embopress.org/page/journal/17444292/authorguide#expandedview>.

- Before submitting your revision, primary datasets (and computer code, where appropriate) produced in this study need to be deposited in an appropriate public database (see <http://msb.embopress.org/authorguide-dataavailability>).

<https://www.embopress.org/page/journal/17444292/authorguide#dataavailability>).

The accession numbers and database should be listed in a formal "Data Availability" section (placed after Materials & Method) that follows the model below (see also <https://www.embopress.org/page/journal/17444292/authorguide#dataavailability>). Please note that the Data Availability Section is restricted to new primary data that are part of this study.

Data availability

- RNA-Seq data: Gene Expression Omnibus GSE46843 (<https://www.ncbi.nlm.nih.gov/geo/query/acc.cgi?acc=GSE46843>)

- [data type]: [name of the resource] [accession number/identifier/doi] ([URL or identifiers.org/DATABASE:ACCESSION])

- At EMBO Press we ask authors to provide source data for the main figures. Our source data coordinator will contact you to

discuss which figure panels we would need source data for and will also provide you with helpful tips on how to upload and organize the files.

- Our journal encourages inclusion of *data citations in the reference list* to directly cite datasets that were re-used and obtained from public databases. Data citations in the article text are distinct from normal bibliographical citations and should directly link to the database records from which the data can be accessed. In the main text, data citations are formatted as follows: "Data ref: Smith et al, 2001". In the Reference list, data citations must be labeled with "[DATASET]". A data reference must provide the database name, accession number/identifiers and a resolvable link to the landing page from which the data can be accessed at the end of the reference. Further instructions are available at .

- We updated our journal's competing interests policy in January 2022 and request authors to consider both actual and perceived competing interests. Please review the policy <https://www.embopress.org/competing-interests> and update your competing interests if necessary. Please use the heading "Disclosure statement and competing interests".

- All Materials and Methods need to be described in the main text using our 'Structured Methods' format. According to this format, the Methods section includes a Reagents and Tools Table (listing key reagents, experimental models, software and relevant equipment and including their sources and relevant identifiers) followed by a Methods and Protocols section describing the methods, ideally using a step-by-step protocol format. The aim is to facilitate adoption of the methodologies across labs. Please download and fill our Reagents and Tools Table template (.docx), which you can find in our author guidelines: <https://www.embopress.org/page/journal/17444292/authorguide#structuredmethods>.

An example of a Method paper with Structured Methods can be found here:
<https://www.embopress.org/doi/10.15252/msb.20178071>.

-Regarding data quantification:

Please ensure to specify the name of the statistical test used to generate error bars and P values, the number (n) of independent experiments (please specify technical or biological replicates) underlying each data point and the test used to calculate p-values in each figure legend. Discussion of statistical methodology can be reported in the materials and methods section, but figure legends should contain a basic description of n, P and the test applied. Graphs must include a description of the bars and the error bars (s.d., s.e.m.). Please also include scale bars in all microscopy images.

- Please provide a "standfirst text" summarizing the study in one or two sentences (approximately 250 characters, including space), three to four "bullet points" highlighting the main findings and a "synopsis image" (550px width and 400-600 px height, PNG format) to highlight the paper on our homepage.

Here are a couple of examples:

<https://www.embopress.org/doi/10.15252/msb.20199356>

<https://www.embopress.org/doi/10.15252/msb.20209475>

<https://www.embopress.org/doi/10.15252/msb.209495>

When you resubmit your manuscript, please download our CHECKLIST (<https://www.embopress.org/pb-assets/embo-site/EMBO%20Press%20Author%20Checklist-1642513524327.xlsx>) and include the completed form in your submission.

Please note that the Author Checklist will be published alongside the paper as part of the transparent process (<https://www.embopress.org/page/journal/17444292/authorguide#transparentprocess>).

If you feel you can satisfactorily deal with these points and those listed by the referees, you may wish to submit a revised version of your manuscript. Please attach a covering letter giving details of the way in which you have handled each of the points raised by the referees. A revised manuscript will be once again subject to review and you probably understand that we can give you no guarantee at this stage that the eventual outcome will be favorable.

I look forward to receiving a revised manuscript soon.

Kind regards,
Jingyi

Jingyi Hou, PhD
Scientific Editor
Molecular Systems Biology

We realize that it is difficult to revise to a specific deadline. In the interest of protecting the conceptual advance provided by the work, we recommend a revision within 3 months (29th Nov 2024). Please discuss the revision progress ahead of this time with the editor if you require more time to complete the revisions. Use the link below to submit your revision:

IMPORTANT: When you send your revision, we will require the following items:

1. the manuscript text in LaTeX, RTF or MS Word format
2. a letter with a detailed description of the changes made in response to the referees. Please specify clearly the exact places in the text (pages and paragraphs) where each change has been made in response to each specific comment given
3. three to four 'bullet points' highlighting the main findings of your study
4. a short 'blurb' text summarizing in two sentences the study (max. 250 characters)
5. a 'thumbnail image' (550px width and max 400px height, Illustrator, PowerPoint or jpeg format), which can be used as 'visual title' for the synopsis section of your paper.
6. Please include an author contributions statement after the Acknowledgements section (see <https://www.embopress.org/page/journal/17444292/authorguide>)
7. Please complete the CHECKLIST available at (<https://bit.ly/EMBOPressAuthorChecklist>).

Please note that the Author Checklist will be published alongside the paper as part of the transparent process (<https://www.embopress.org/page/journal/17444292/authorguide#transparentprocess>).

See also figure legend guidelines: <https://www.embopress.org/page/journal/17444292/authorguide#figureformat>

9. Please note that corresponding authors are required to supply an ORCID ID for their name upon submission of a revised manuscript (EMBO Press signed a joint statement to encourage ORCID adoption). (<https://www.embopress.org/page/journal/17444292/authorguide#editorialprocess>)

Currently, our records indicate that the ORCID for your account is 0000-0002-5607-3845.

Link Not Available

11. Include a Reagents and Tools Table as part of the Methods section, which can be downloaded from our author guidelines (<https://www.embopress.org/page/journal/17444292/authorguide#structuredmethods>)

*** PLEASE NOTE *** As part of the EMBO Press transparent editorial process initiative (see our Editorial at <https://dx.doi.org/10.1038/msb.2010.72>), Molecular Systems Biology publishes online a Review Process File with each accepted manuscripts. This file will be published in conjunction with your paper and will include the anonymous referee reports, your point-by-point response and all pertinent correspondence relating to the manuscript. If you do NOT want this File to be published, please inform the editorial office at msb@embo.org within 14 days upon receipt of the present letter.

Rev_Com_number: RC-2024-02622

New_manu_number: MSB-2024-12724-T

Corr_author: Hoffmann

Title: Synergy and antagonism in the integration of BCR and CD40 signals that control B-cell proliferation

Reviewer #1 (Evidence, reproducibility and clarity (Required)):

Summary:

This manuscript uses a mathematical model to explore integration of B cell receptor (BCR) and CD40 stimulation of B cells and how it affects the signaling and proliferation response. The authors found that to accurately model B cell responses to co-stimulation required connecting CD40-induced signaling to a proliferation module and BCR-induced signaling to an activation induced cell death module. The authors showed that BCR stimulation synergizes with CD40 signaling at low doses to amplify proliferation but at high doses, the BCR induces apoptosis in founder cells that antagonizes proliferation induced by CD40. The interaction between these modules is time-dependent, which the author suggest can tune the positive and negative selection of B cells. The conclusions are supported by a mathematical model that is well described and with code available. Many key predictions of the model are tested with experimental data.

We thank the reviewer for their careful reading of the paper and their appreciation of the computational model and the experimental testing of predictions.

Major comments:

- Fig. 4: A major conclusion of this figure is that "NF- κ B signaling saturation explains the dose-dependent interaction in costimulation" however Fig. 4F does not have experimental validation showing that RelA and cRel activity stimulated by low-dose α -CD40 are in fact amplified by α -IgM at high dose as the model predicts. It seems important to include this data point, especially since the high dose experimental points are rather noisy.
- Regarding the immunoblot experiments, there are no error bars included for any of the immunoblot data. Were these done with replicates?

We agree with the reviewer that additional datasets would be helpful in further informing or testing the model. Rather than undertaking more immunoblots that represent poorly quantifiable population averages, we conducted a single-cell resolution experimental approach using B-cells from a new RelA and cRel double fluorescent reporter mouse (Vaidehi Narayanan et al. *PNAS* 2024). This allowed us to obtain distributions of RelA and cRel activity as well as their time-dependent evolution. The single cell resolution data revealed that there is wide cell heterogeneity in the expression of these regulators, as shown in Fig. 4F and H, consistent with model prediction. While these experiments are the key reason for the long delay in the resubmission of the manuscript, we feel that they are a significant improvement over the immunoblots in terms of single-cell resolution, and also showed good consistency with the immunoblot quantification.

- Fig. 7: Another key conclusion is that noise in BCR-induced Bcl-xL expression, as well as NF- κ B activity, determines the balance of apoptosis and proliferation in cell populations. The simulations in B and C predict levels of heterogeneity in key signaling molecules. Testing this experimentally using flow cytometry for some of the molecules (e.g., I κ B α as a proxy for RelA activity and caspase 8) would strengthen these predictions.

We thank the reviewer for this suggestion. We apologize for not citing the relevant papers that show that Bcl-xL is a well-established RelA and cRel target gene and anti-apoptotic regulator (Chen et al., *Mol Cell Biol* 2000). Previous literature, using fluorescence microscopy and qPCR, have found a significant reduction in Bcl-xL protein and mRNA expression in CpG-stimulated B-cells absent of cRel (Shokhirev et al., *MSB* 2015, Fig. 4J-K, S5B). Furthermore, Berry et al. (*Cell Reports*, 2020) found that BCR-induced Bcl-xL level had a much wider distribution than basal, with only part of the population being upregulated (Fig. 3B, 3G). They also found a moderately positive correlation between cRel and Bcl-xL expression with a Pearson coefficient of 0.48 (Fig. 3G). As cRel is subject to an autoregulated loop, in (quasi) steady-state cRel expression is indicative of cRel nuclear activity.

Minor comments:

- Line 30: "co-stimulation with α -CD40 and α -BCR". The authors refer to α -IgM rather than α -BCR for the rest of the paper and so it's confusing to write it differently here.

We thank the reviewer for pointing this out. The nomenclature was indeed confusing when we use α -BCR and α -IgM interchangeably. Because we focus on the signaling function of the BCR, we have decided to change all the mentions of " α -IgM" to " α -BCR" in the paper. We added that " α -BCR" referred to " α -IgM" in the Materials and Methods (B cell isolation and culture).

- Fig. 6C: it looks like some smoothing was done on these experimental curves but I couldn't find where the method was described.

We apologize for missing the description for this smoothed line. We have added description of the smoothed line to the figure caption.

Reviewer #1 (Significance (Required)):

A strength of this study is the use of a mathematical model to consider complex questions of timing and signal integration that are difficult to thoroughly explore

experimentally due to time/cost and difficulty of interpreting of the results. The authors show that the model provides biologically relevant insights and include some experimental validation, which is also a strength. A limitation is that some of the more complex predictions are difficult to experimentally validate. However, it would improve the study and likely be more convincing to non-mathematical immunologists to expand some of this validation.

Advance: In systems biology, this study is a novel example of integrating mathematical models at different time scales, and understanding mechanisms underlying positive and negative selection is an important basic research area. I do not study B cell biology, and so I'm less familiar with how it fits into the current literature in that field.

Audience: The audience for this paper would be both systems biologists integrating modeling and experiment to study key signaling pathways, as well as B cell immunologists interested in this specific application area.

We appreciate the assessment. We agree that the work will be of interest to both systems biologists and B-cell immunologists. Computational models are of increasing importance and relevance to biological studies and the current work documents a significant advance in how B-cells integrate the two physiologically relevant signals from the B-cell receptor (antigen) and the CD40 receptor (T-cells). We have added additional experimental work, as described, to ensure that experimentally trained biologists have confidence in the insights generated by the computational model which represents an integration of prior knowledge of experimental biology studies.

Reviewer #2 (Evidence, reproducibility and clarity (Required)):

Summary:

The study investigates how BCR and CD40 signaling influence B-cell fate during the humoral immune response. Using a differential-equations-based model of NF- κ B activation, the model accurately predicts NF- κ B dynamics following co-stimulation, but initially overestimates B-cell population expansion compared to stimulation experiments. This discrepancy is attributed to BCR-induced caspase activity leading to apoptosis unless counteracted by timely NF- κ B-driven survival signals. Sequential co-stimulation experiments confirm that the temporal dynamics of BCR and CD40 signaling determine the balance between positive and negative B-cell selection during T-cell activation.

We thank the reviewer for the careful reading of the manuscript.

Major:

- The authors need to more clearly distinguish between naive B cell activation by antigens and T-cells, and the affinity maturation process of centrocytes in the germinal center (GC), especially in the introduction and discussion. Although the study focused solely on naive B-cell activation, the findings were extrapolated to centrocytes affinity maturation in several instances (line 504, 561). While the term "founder cells" was used, the distinction should be made clearer, particularly in the discussion.

We apologize for the confusion between naïve B cells and GC B cells in the introduction and discussion. The reviewer's comments provide a helpful roadmap for the fix. Our work builds on prior literature on experimental models of T-dependent naïve B-cell activation which are mentioned in the introduction. We then extrapolate our findings in the 5th paragraph of Discussion, when we mention the germinal center and how the known GC B-cell phenotype affects the reliability of extrapolating the findings.

- Given the similarities between the NF- κ B pathway induced by antigens in founder cells and by follicular dendritic cells (FDCs) in centrocytes, the authors could broaden the impact of their findings by exploring or discussing how their findings might also apply to centrocyte affinity maturation. However, before delving into this, it is crucial to clearly explain the differences between FDC peptide uptake and antigen activation in the context of the NF- κ B pathway.

We agree with the reviewer that it's crucial to explain the differences between pulling antigen from an FDC in centrocytes vs. directly recognizing antigens in pre-GC B-cells. We have included clear description of this difference in the 5th paragraph of discussion: Pre-GC B-cells are naive B-cells that recognize soluble or membrane-bound antigens through their BCR, typically in lymphoid tissue before the formation of germinal center. This BCR engagement directly activates the NF κ B pathway through a series of adapter proteins. On the other hand, centrocytes in the GC sample and uptake antigens that are presented in immune complexes on the surface of follicular dendritic cells (FDCs) through their BCR. It has been shown that centrocytes downregulate their BCR-induced NF κ B activation compared to early activated B-cells (Young and Brink, 2021).

Minor:

- Figure 1: The title, "Multi-scale model recapitulates B-cell NF κ B dynamics in response to T-dependent stimulation," is misleading, as the model does not accurately capture the dynamics at later time points. Additionally, it cannot be considered truly multiscale since it omits key processes such as B-cell division and death. The title should be revised to better reflect the figure.

We thank the reviewer for pointing out this important point. The model presented in Figure 1 is indeed not multi-scale – only the models in later figures are. We have

revised the title to more accurately reflect this fact and the results by removing “multi-scale” and adding “early”.

- Figure 1A: Since follicular dendritic cells (FDCs) are not involved in the activation of founder cells but only in the selection of centrocytes, they should be removed from the figure.

We thank the reviewer for pointing out this discrepancy in the figure and the biology we are presenting so we have removed the FDCs and T cells from the figure to avoid confusion.

- Figure 1A: The description, "Schematic of BCR-CD40 receptor model to recapitulate T-dependent activation of B-cells," needs more detail. What do the numbers represent? What do the circles with a red 'S' signify? These elements should be clarified in the figure's explanation.

We apologize for the lack of detail. In the revision we have added explanations for each of the elements in the caption and revised the figure to include an expanded legend. For example, the circles with a red 'S' indicates a hill activation and it is labeled in the legend in the figure now.

- Figure 1C,D: What do A:50, A:52, C:50, and C:52 represent? Please provide an explanation for these notations in the Figure's caption.

We apologize for not explaining the short hand: these are the NF κ B dimers: RelA:p50, cRel:p50 etc. In the revision we have now added explanations in the figure caption.

- Figures 3 and 4: The "hi" and "low" labels should be reversed in one of the figures to ensure consistency between the two.

We thank the reviewer for pointing this out. To maintain visual consistency between figure 3 and 4, we have switched the panels in figure 4.

- Figure 4: A more detailed schematic of caspase-8 (including PC8, C8, and ABCR) would be helpful, as this is the authors' main contribution to explain the experimental data.

We agree with the reviewer that this is an important figure. We have now added a more detailed schematic of caspase-8 processing that involved PC8, C8, and ABCR to Fig. 4A. The kinetic constant was added to the supplement parameter table mentioned in the next comment.

- Table 1: The table currently only covers the model described in section one. It would

be beneficial to add information about the complete multiscale model, highlighting the number of parameters from other pathways and the newly introduced caspase-8 pathway. Either expand Table 1 or create a new table to include this additional information.

We agree with the reviewer about the importance of a master parameter table for the complete multiscale model, and apologize for the oversight. We have presented this table in the supplement.

Reviewer #2 (Significance (Required)):

This work shed light into the complex mechanism of founder B cell activation through antigen and T-cell co-stimulation, providing a systematic study that includes experimental validation to refine and enhance existing gene regulatory networks (GRNs). This work is particularly significant as it offers a detailed quantitative assessment of a previously unknown co-stimulation in BCR activation, supported by experimental evidence and convincing quantitative measures.

We thank the reviewer for appreciating the significance of providing a detailed model of known molecular mechanisms to explore B-cell responses to stimulation, and the experimental support for this effort.

However, the study's use of a highly complex system with several hundreds of parameters raises concerns about the feasibility of integrating this GRN into a full scale germinal center mechanistic model [such as doi.org/10.1038/s41540-023-00271-y, <https://doi.org/10.3390/cells9061448>]. Simplifying the model by focusing on a few critical interactions and parameters could be beneficial to the community, as it has been previously done for example with Memory vs. Plasma blast B-cell differentiation (doi.org/10.1073/pnas.111301910), which utilized only a dozen of reactions. Moreover, the study's focus on Naive B cell activation, without addressing the role of centrocytes during the germinal center reaction, limits its scope to B cell activation rather than the broader process of B cell affinity maturation.

We thank the review for raising concerns of using a highly complex model with several hundreds of parameters. We'd like to address two aspects to this:

1) It is important to distinguish model constructions that attempt to produce highly complex model *de novo*, versus research in which complex models are the result of a decades long process of iterative construction. That is the case in the present model. For example, the NFκB signaling model has been parameterized extensively with biochemical and biophysical studies between 2002 and 2012 and recently was shown to capture the NFκB system regulation in a variety of cell types and stimulus conditions (Mitchell et al 2023). Similar effector modules of apoptosis and cell cycle have

undergone rigorous analysis in the groups of Peter Sorger and John Tyson, and our own lab has vetted these for B-cells in four prior publications (MSB 2015, PNAS 2018, Immunity 2019, PNAS 2024). Therefore, the present model represents a (growing) knowledge base of the molecular mechanisms that govern B-cell responses to stimulation.

2) We agree with the reviewer that in order to simulate B-cell responses at the organ level or organism level, the present level of detail may be too cumbersome, and may be computationally too expensive. We thank the reviewer for pointing out some papers that have presented elegant models with fewer parameters that address the organ level outcomes of the germinal center reaction in terms of affinity maturation. However, these models do not contain the molecular mechanisms that may be affected by genetic variants or may potentially be targeted pharmacologically. We agree with the reviewer that the detailed model we present would need to be carefully pared down to preserve molecular mechanisms of interest while rendering the simulations less computationally expensive.

Despite these limitations, the discovery of a new potential mechanism represents a significant advancement in the field. The paper is likely to appeal to a specialized audience, particularly those familiar with mechanistic models and the pathways involved in the B-cell cycle, as well as T-cell and antigen interactions.

We are pleased that the reviewer appreciated that we report a significant advance for the field, as we reveal key mechanisms that govern the stringency of B-cell selection. It shows how negative and positive selection are intertwined, with BCR mediating negative selection, while enabling positive selection that comes from T-cells via CD40.

29th Apr 2025

Manuscript Number: MSB-2024-12724R

Title: Synergy and antagonism in the integration of BCR and CD40 signals that control B-cell proliferation

Author: Helen Huang

Haripriya Vaidehi Narayanan

Mark Xiang

Vaibhava Kesarwani

Alexander Hoffmann

Dear Alex,

Thank you for submitting your revised manuscript to Molecular Systems Biology. We have now received the enclosed reports from two reviewers who agreed to re-assess your work. As you will see below, both reviewers are overall satisfied with the revisions. Therefore, I am pleased to inform you that we will be able to accept your manuscript pending the following amendments:

1. Please address the remaining comment from Reviewer #1 (formerly Reviewer #2 in the previous review round).

On a more editorial level, we kindly ask that you address the following issues:

1. Please label both corresponding authors on the title page and include their email addresses.

2. Please remove the "Authors' Contribution" section from the manuscript file.

3. Please add the missing information related to n is missing in the legends of Figures 4F, H.

4. In the "Disclosure and competing interests statement", please include the following - Alexander Hoffmann is a member of the Advisory Editorial Board of Molecular Systems Biology. This has no bearing on the editorial consideration of this article for publication.

5. Please remove the "standfirst and highlights" from the manuscript file and upload them as a separate file.

6. Callouts:

- Please ensure that all callouts are listed sequentially.
- Add callouts for Dataset EV1, Figure 5L and Figure 5N.
- Correct the callout "5I-K" to "Figure 5I-K".

7. Source file names, titles, legends and manuscript callouts all need to be updated to Dataset EV2 instead of Table EV1. The legend should be included as a separate tab/sheet in the same Excel file.

8. Please download and fill our Reagents and Tools Table template (.docx), which you can find in our author guidelines: <https://www.embopress.org/page/journal/17444292/authorguide#structuredmethods>.

9. Please correct the section order to the following: Title page - Abstract & Keywords - Introduction - Results - Discussion - Methods - Data Availability - Acknowledgements - Disclosure and Competing Interests Statement - References - Figure Legends - Table(s) - Expanded View Figure Legends.

Please resubmit your revised manuscript online, with a covering letter listing amendments and responses to each point raised by the referees. Please resubmit the paper ****within one month**** and ideally as soon as possible. Please use the Manuscript Number (above) in all correspondence.

Click on the link below to submit your revised paper.

Kind regards,
Jingyi

Jingyi Hou, PhD
Senior Editor
Molecular Systems Biology

*** PLEASE NOTE *** As part of the EMBO Press transparent editorial process initiative (see our Editorial at <https://dx.doi.org/10.1038/msb.2010.72> , Molecular Systems Biology will publish online a Review Process File to accompany accepted manuscripts. When preparing your letter of response, please be aware that in the event of acceptance, your cover letter/point-by-point document will be included as part of this File, which will be available to the scientific community. More information about this initiative is available in our Instructions to Authors. If you have any questions about this initiative, please contact the editorial office (msb@embo.org).

Reviewer #1:

The reviewers have thoroughly addressed all of my previous comments, which has significantly strengthened the manuscript. I do not have any further major concerns.

I do, however, have one additional suggestion for clarification:

Lines 174-175:

"Inspecting the data, we found that increasing doses of CD40 affect both the time to first division (T_{div0}) and the total number of divisions a B-cell can reach."

Could the authors expand more quantitatively on the notion of "number of divisions" in B cell costimulation with BCR and CD40? The figures suggest that costimulation can lead to up to 7 cell divisions, which may have important implications for the strength and duration of an immune response, as the B-cell progeny can seed germinal centers or further differentiate into effector cells.

- Is 7 the observed maximum? And does the stimulatory effect plateau or diminish at higher levels of stimulation?
- Does the number of divisions increase gradually with stronger BCR and CD40 signaling inputs, or does the system behave more like a thresholded response, where cells either don't divide or commit to multiple divisions (e.g., 5+) once a certain signal strength is reached??

A heatmap in the main or supplementary figures, showing predicted division number as a function of BCR and CD40 co-stimulation strength, could provide a helpful visualization of how input signals are integrated to shape the proliferative response.

Earlier work (e.g., Hasbold et al., 1998: <https://pubmed.ncbi.nlm.nih.gov/9541600/>); and subsequent studies (Turner et al., 2008: <https://doi.org/10.4049/jimmunol.181.1.374>) suggest a graded response that can reach up to 8 divisions. However, in an in vivo context, how frequently are the required thresholds for CD40 and BCR signaling met to drive cells to the maximum number of divisions? Does this occur via a slow, progressive accumulation of signals or through a more abrupt, switch-like mechanism? A brief discussion of these aspects would enhance the reader's understanding of how signal strength translates into functional B cell proliferation and downstream responses.

Reviewer #2:

This manuscript explores integration of B cell receptor (BCR) and CD40 stimulation of B cells with a mechanistic mathematical model to predict how it will affect the signaling and proliferation response of B cells. To accurately model B cell responses to co-stimulation required connecting CD40-induced signaling to a proliferation module and BCR-induced signaling to an activation induced cell death (AICD) module. The authors showed that BCR stimulation synergizes with CD40 signaling at low doses to amplify proliferation but at high doses, the BCR induces apoptosis in founder cells that antagonizes proliferation induced by CD40. The interaction between these modules is time-dependent--BCR-induced caspase activity will lead to apoptosis unless counteracted by CD40-induced NF- κ B-driven survival signals, mimicking the physiological scenario and providing novel insights.

This is a significant finding that demonstrates how models calibrated over many studies over time can be reliably connected to make biologically accurate predictions. The authors validate novel predictions experimentally, including the model prediction of differential signaling saturation in the NF- κ B pathway. The model is well described, with assumptions clear and code available.

Based on my reading of the other reviewer's comments who is in the B-cell field, the biological discovery is novel and of interest to these experts.

I previously reviewed this paper with two major comments:

1. One of the key findings of this paper is that BCR stimulation synergizes with low-dose CD40 but antagonizes high-dose CD40 stimulation. The signaling model predicted that this was due to synergy in signaling at the low dose while signaling saturation was occurring when BCR was combined with a high dose of CD40. In this revision, the authors have experimentally validated this with a live cell RelA-cRel-H2B reporter that allowed them to quantitatively assess signaling activation under each of the relevant conditions. These experiments provide strong support for model predictions and the conclusions.

2. The other comment was regarding experimental validation of the prediction that as the timing of CD40 post BCR stimulation is extended, AICD via caspase 8 is induced in a larger fraction of the population and that this leads to the attenuation of proliferation. In silico, the authors trace the heterogeneity back to differences in Rel-A and cRel activity, which are upstream TFs for Bcl-xL, which means that cells with higher activity have more protection against apoptosis.

- a. This is an interesting prediction about the functional consequences of cell-cell heterogeneity. It would be difficult to validate experimentally and thus I think such an experiment is outside the scope of this paper.
- b. My comment/suggestion was to experimentally confirm with flow cytometry that the fraction of cells with cleaved caspase 8 increases as predicted when the CD40ag window is extended. Confirming of AICD via activation of caspase 8 is the only arm of this model that is not shown experimentally, and it would emphasize the heterogeneous nature of the individual cell outcomes as predicted. However, I consider this a minor point and not critical for publication, as the authors have already provided sufficient validation for their model predictions.

Reviewer #1:

The reviewers have thoroughly addressed all of my previous comments, which has significantly strengthened the manuscript. I do not have any further major concerns.

We thank the reviewer for their assessment and previous suggestions that have improved the paper.

I do, however, have one additional suggestion for clarification:

Lines 174-175:

"Inspecting the data, we found that increasing doses of CD40 affect both the time to first division (Tdiv0) and the total number of divisions a B-cell can reach."

Could the authors expand more quantitatively on the notion of "number of divisions" in B cell costimulation with BCR and CD40? The figures suggest that costimulation can lead to up to 7 cell divisions, which may have important implications for the strength and duration of an immune response, as the B-cell progeny can seed germinal centers or further differentiate into effector cells.

- Is 7 the observed maximum? And does the stimulatory effect plateau or diminish at higher levels of stimulation?

We thank the reviewer for the suggestion. In reviewing the evidence, we must acknowledge the limitations of the *in vitro* experimental system as well as the mathematical model in characterizing the number of divisions that stimulated B-cells will undergo. The experimental system relies on dye dilution after unstimulated cells have been loaded up with a tracing dye. While the resolution can be pushed to 7 or 8 divisions, when dye drops to <1% the reliability drops, see for example, Hasbold et al., 1998, Figure 1 top left panel.

Secondly, prolonged proliferative culture is subject to nutrient depletion, which may not reflect the true *in vivo* metabolic exhaustion that may limit proliferative capacity *in vivo*. Therefore, while the experimental system can address speed of response and total cell count within the given constraints, total number of generations may not be reliable. We have revised the methods to indicate this and described the limitations.

- Does the number of divisions increase gradually with stronger BCR and CD40 signaling inputs, or does the system behave more like a thresholded response, where cells either don't divide or commit to multiple divisions (e.g., 5+) once a certain signal strength is reached?

Our observation that increasing doses of CD40 affect both Tdiv0 and the total number of divisions a B-cell can reach is qualitatively consistent with previous reports of a graded response to CD40 stimulation versus a quantal all-or-none response to CpG and LPS. We have added this clarification in line 162-163 and cited the Turner et al. 2008 paper you suggested, and a follow-up Hawkins et al. 2013 paper (<https://www.nature.com/articles/ncomms3406>). However, we are unable to make reliable statements about the total number of divisions for the reasons indicated above.

A heatmap in the main or supplementary figures, showing predicted division number as a function of BCR and CD40 co-stimulation strength, could provide a helpful visualization of how input signals are integrated to shape the proliferative response.

This is also a nice idea, but unfortunately we are unable to provide reliable information about the number of divisions. The mathematical model simulations cap the maximum number of divisions to match the experimental results, which are not entirely reliable as discussed above. We do not presently know the precise molecular mechanism that caps the division number and so this exceeds the capability of the present model. Furthermore, division destiny is a

distribution among many cells, rather than a single number for each condition. We are not sure a heatmap of maximum division number would be helpful, as it is not representative of the whole proliferation picture. For example, we observed up to 6 divisions even in low CD40 dose, but most cells did not divide even once, and much fewer cells were able to divide 6 times in low CD40 than in high CD40 dose. Therefore, quantifying the percentage of cells that divided as well as the population size fold change is more informative of the proliferative response.

Earlier work (e.g., Hasbold et al., 1998: <https://pubmed.ncbi.nlm.nih.gov/9541600/>); and subsequent studies (Turner et al., 2008: <https://doi.org/10.4049/jimmunol.181.1.374>) suggest a graded response that can reach up to 8 divisions. However, in an in vivo context, how frequently are the required thresholds for CD40 and BCR signaling met to drive cells to the maximum number of divisions? Does this occur via a slow, progressive accumulation of signals or through a more abrupt, switch-like mechanism? A brief discussion of these aspects would enhance the reader's understanding of how signal strength translates into functional B cell proliferation and downstream responses.

We thank the reviewer for the thoughtful discussion points. Due to the need to either differentiate to plasma cell or recycle to the dark zone, in vivo GC B-cells may not reach the same division destiny in a single GC cycle as in vitro. In the in vivo context, a study has shown that pre-GC B-cells undergo 4-6 divisions after the first round of stimulation before differentiating into GC B-cells (Zhang et al., 2017). Another in vivo study investigated the effects of B-cell competition on proliferation in early GC and found 88% low-affinity B-cells divided without competition, while only 29% divided with a competing high-affinity B-cell clone; competition did not affect antigen binding (BCR signal) or presentation, but rather affected the access to T-cell help (CD40 signal) (Figure 2B, Schwickert et al., 2011). While they used CFSE dye dilution stain, they did not quantify the number of divisions, but it's clear that the average number of divisions was reduced under competition. This indicated that the frequency at which CD40 and BCR signaling met the maximum thresholds depends on the competition level, B-cell affinity, and the access to T-cells.

BCR signaling was commonly believed to be transient in vivo (Damdinsuren et al. 2010). T-cell help was shown to be mediated by large but transient contacts between GC B-cells and cognate Tfh (Shulman 2014). However, it's unclear whether accumulation of signals is needed or not.

Reviewer #2:

This manuscript explores integration of B cell receptor (BCR) and CD40 stimulation of B cells with a mechanistic mathematical model to predict how it will affect the signaling and proliferation response of B cells. To accurately model B cell responses to co-stimulation required connecting CD40-induced signaling to a proliferation module and BCR-induced signaling to an activation induced cell death (AICD) module. The authors showed that BCR stimulation synergizes with CD40 signaling at low doses to amplify proliferation but at high doses, the BCR induces apoptosis in founder cells that antagonizes proliferation induced by CD40. The interaction between these modules is time-dependent--BCR-induced caspase activity will lead to apoptosis unless counteracted by CD40-induced NF- κ B-driven survival signals, mimicking the physiological scenario and providing novel insights.

This is a significant finding that demonstrates how models calibrated over many studies over time can be reliably connected to make biologically accurate predictions. The authors validate novel predictions experimentally, including the model prediction of differential signaling saturation in the NF- κ B pathway. The model is well described, with assumptions clear and code available. Based on my reading of the other reviewer's comments who is in the B-cell field, the biological discovery is novel and of interest to these experts.

I previously reviewed this paper with two major comments:

1. One of the key findings of this paper is that BCR stimulation synergizes with low-dose CD40 but antagonizes high-dose CD40 stimulation. The signaling model predicted that this was due to synergy in signaling at the low dose while signaling saturation was occurring when BCR was combined with a high dose of CD40. In this revision, the authors have experimentally validated this with a live cell RelA-cRel-H2B reporter that allowed them to quantitatively assess signaling activation under each of the relevant conditions. These experiments provide strong support for model predictions and the conclusions.

2. The other comment was regarding experimental validation of the prediction that as the timing of CD40 post BCR stimulation is extended, AICD via caspase 8 is induced in a larger fraction of the population and that this leads to the attenuation of proliferation. In silico, the authors trace the heterogeneity back to differences in Rel-A and cRel activity, which are upstream TFs for Bcl-xL, which means that cells with higher activity have more protection against apoptosis.

a. This is an interesting prediction about the functional consequences of cell-cell heterogeneity. It would be difficult to validate experimentally and thus I think such an experiment is outside the scope of this paper.

b. My comment/suggestion was to experimentally confirm with flow cytometry that the fraction of cells with cleaved caspase 8 increases as predicted when the CD40ag window is extended. Confirming of AICD via activation of caspase 8 is the only arm of this model that is not shown experimentally, and it would emphasize the heterogeneous nature of the individual cell outcomes as predicted. However, I consider this a minor point and not critical for publication, as the authors have already provided sufficient validation for their model predictions.

We thank the reviewer for their assessment and previous suggestions that have improved the paper.

19th May 2025

Manuscript number: MSB-2024-12724RR

Title: Synergy and antagonism in the integration of BCR and CD40 signals that control B-cell proliferation

Dear Alex,

Thank you again for sending us your revised manuscript. We are now satisfied with the modifications made and I am pleased to inform you that your paper has been accepted for publication.

Kind regards,
Jingyi

Jingyi Hou, PhD
Senior Editor
Molecular Systems Biology

Rev_Com_number: RC-2024-02622
New_manu_number: MSB-2024-12724RR
Corr_author: Hoffmann
Title: Synergy and antagonism in the integration of BCR and CD40 signals that control B-cell proliferation